


**Historical and future contributions of inland waters to the Congo basin**
**carbon balance**
Adam Hastie[1,2], Ronny Lauerwald[2,3], Philippe Ciais[3], Fabrice Papa[4,5], Pierre Regnier[2]
[1]School of GeoSciences, University of Edinburgh, EH9 3FF, Edinburgh, Scotland, UK
[2]Biogeochemistry and Earth System Modelling, Department of Geoscience, Environment and
Society, Universite Libre de Bruxelles, Bruxelles, 1050, Belgium
[3]Laboratoire des Sciences du Climat et de l'Environnement (LSCE), CEA CNRS UVSQ, Gif-
sur-Yvette 91191, France
[4]Laboratoire d'Etudes en Géophysique et Océanographie Spatiales, Centre National de la
Recherche Scientifique–Institut de recherche pour le développement–Université Toulouse Paul
Sabatier–Centre national d'études spatiales, 31400 Toulouse, France
[5]Indo-French Cell for Water Sciences, International Joint Laboratory Institut de Recherche
pour le Développement and Indian Institute of Science, Indian Institute of Science, 560012
Bangalore, India
*Correspondence to*: Adam Hastie (adam.hastie@ed.ac.uk)
**Abstract**
As the second largest area of contiguous tropical rainforest and second largest river basin in
the world, the Congo basin has a significant role to play in the global carbon (C) cycle.
Inventories suggest that terrestrial net primary productivity (NPP) and C storage in tree biomass
has increased in recent decades in intact forests of tropical Africa, due in large part to a
combination of increasing atmospheric $CO_2$ concentrations and climate change, while
rotational agriculture and logging have caused C losses. For the present day, it has been shown
that a significant proportion of global terrestrial NPP is transferred laterally to the land-ocean
aquatic continuum (LOAC) as dissolved $CO_2$, dissolved organic carbon (DOC) and particulate
organic carbon (POC). Whilst the importance of LOAC fluxes in the Congo basin has been
demonstrated for the present day, it is not known to what extent these fluxes have been
perturbed historically, how they are likely to change under future climate change and land use



scenarios, and in turn what impact these changes might have on the overall C cycle of the basin.
Here we apply the ORCHILEAK model to the Congo basin and show that 4% of terrestrial
NPP (NPP = 5,800 ±166 Tg C yr⁻¹) is currently exported from soils to inland waters. Further,
we found that aquatic C fluxes have undergone considerable perturbation since 1861 to the
present day, with aquatic $CO_2$ evasion and C export to the coast increasing by 26% (186 ±41
Tg C yr⁻¹ to 235 ±54 Tg C yr⁻¹) and 25% (12 ±3 Tg C yr⁻¹ to 15 ±4 Tg C yr⁻¹) respectively,
largely because of rising atmospheric $CO_2$ concentrations.  Moreover, under climate scenario
RCP 6.0 we predict that this perturbation will continue; over the full simulation period (1861-
2099), we estimate that aquatic $CO_2$ evasion and C export to the coast will increase by 79%
and 67% respectively. Finally, we show that the proportion of terrestrial NPP lost to the LOAC
also increases from approximately 3% to 5% from 1861-2099 as a result of increasing
atmospheric $CO_2$ concentrations and climate change.
**1.  Introduction**
As the world's second largest area of contiguous tropical rainforest and second largest river,
the Congo basin has a significant role to play in the global carbon (C) cycle. Approximately 50
Pg C is stored in its above ground biomass (Verhegghen et al., 2012), and up to 100 Pg C
contained within its soils (Williams et al., 2007). Moreover, a recent study estimated that
around 30 Pg C is stored in the peats of the Congo alone (Dargie at al., 2017). Field data suggest
that storage in tree biomass increased by 0.34 Pg C yr⁻¹ in intact African tropical forests
between 1968-2007 (Lewis et al., 2009) due in large part to a combination of increasing
atmospheric $CO_2$ concentrations and climate change (Ciais et al., 2009; Pan et al., 2015), while
satellite data indicates that terrestrial net primary productivity (NPP) has increased by an
average of 10 g C m⁻² yr⁻¹ per year between 2001 and 2013 in tropical Africa (Yin et al., 2017).
At the same time, forest degradation, clearing for rotational agriculture and logging are causing
C losses to the atmosphere (Zhuravleva et al., 2013; Tyukavina et al., 2018) while droughts



have reduced vegetation greenness and water storage over the last decade (Zhou et al., 2014).
A recent estimate of above ground C stocks of tropical African forests, mainly in the Congo,
indicates a minor net C loss from 2010 to 2017 (Fan et al., 2019).
There are large uncertainties associated with projecting future trends in the Congo basin
terrestrial C cycle, firstly related to predicting which trajectories of future $CO_2$ levels and land
use changes will occur, and secondly our ability to fully understand and simulate these changes
and in turn their impacts. Future model projections for the 21st century agree that temperature
will significantly increase under both low and high emission scenarios (Haensler et al., 2013),
while precipitation is only projected to substantially increase under high emission scenarios,
the basin mean remaining more or less unchanged under low emission scenarios (Haensler et
al., 2013). Uncertainties in future land-use change projections for Africa are among the highest
for any continent (Hurtt et al., 2011).
For the present day at global scale, it has been estimated that between 1 and 5 Pg C yr$^{-1}$ is
transferred laterally to the land-ocean aquatic continuum (LOAC) as dissolved $CO_2$, dissolved
organic carbon (DOC) and particulate organic carbon (POC) (Cole at al., 2007; Battin et al.,
2009; Regnier et al., 2013; Drake et al., 2018; Ciais et al. in review). This C can subsequently
be evaded back to the atmosphere as $CO_2$, undergo sedimentation in wetlands and inland
waters, or be transported to estuaries or the coast. The tropical region is a hotspot area for
inland water C cycling (Lauerwald et al., 2015) due to high terrestrial NPP and precipitation,
and a recent study used an upscaling approach based on observations to estimate present day
$CO_2$ evasion from the rivers of the Congo basin at 251±46 Tg C yr$^{-1}$ and the lateral C (TOC
+DIC) export to the coast at 15.5 (13-18) Tg C yr$^{-1}$ (Borges at al., 2015[a]; Borges et al., 2019).
To put this into context, their estimate of aquatic $CO_2$ evasion represents 39% of the global
value estimated by Lauerwald et al. (2015, 650 Tg C yr$^{-1}$) or 14% of the global estimate of
Raymond et al. (2013, 1,800 Tg C yr$^{-1}$).



Whilst the importance of LOAC fluxes in the Congo basin has been demonstrated for the
present day, it is not known to what extent these fluxes have been perturbed historically, how
they are likely to change under future climate change and land use scenarios, and in turn what
impact these changes might have on the overall C balance of the Congo. In light of these
knowledge gaps, we address the following research questions:
• What is the relative contribution of LOAC fluxes ($CO_2$ evasion and C export to the

87       coast) to the present-day C balance of the basin?

• To what extent have LOAC fluxes changed from 1860 to the present day and what are

89       the primary drivers of this change?

• How will these fluxes change under future climate and land use change scenarios (RCP

91       6.0 which represents the "no mitigation scenario") and what are the implications of this

92       change?


Understanding and quantifying these long-term changes requires a complex and integrated
mass-conservation modelling approach. The ORCHILEAK model (Lauerwald et al., 2017), a
new version of the land surface model ORCHIDEE (Krinner et al., 2005), is capable of
simulating both terrestrial and aquatic C fluxes in a consistent manner for the present day in
the Amazon (Lauerwald et al., 2017) and Lena (Bowring et al., 2019[a]; Bowring et al., 2019[b])
basins. Moreover, it was recently demonstrated that this model could recreate observed
seasonal and interannual variation in Amazon aquatic and terrestrial C fluxes (Hastie et al.,

101   2019).

In order to accurately simulate aquatic C fluxes, it is crucial to provide a realistic representation
of the hydrological dynamics of the Congo River, including its wetlands. Here, we develop
new wetland forcing files for the ORCHILEAK model from the high-resolution dataset of
Gumbricht et al. (2017) and apply the model to the Congo basin.  After validating the model




against observations of discharge, flooded area and DOC concentrations for the present day,
we then use the model to understand and quantify the long- term (1861-2099) temporal trends
in both the terrestrial and aquatic C fluxes of the Congo Basin.
**2.   Methods**
ORCHILEAK (Lauerwald et al., 2017) is a branch of the ORCHIDEE land surface model
(LSM), building on past model developments such as ORCHIDEE-SOM (Camino Serrano,
2015), and represents one of the first LSM-based approaches which fully integrates the aquatic
C cycle within the terrestrial domain.  ORCHILEAK simulates DOC production in the canopy
and soils, the leaching of dissolved $CO_2$ and DOC to the river from the soil, the mineralization
of DOC, and in turn the evasion of $CO_2$ to the atmosphere from the water surface. Moreover,
it represents the transfer of C between litter, soils and water within floodplains and swamps
(see section 2.2). Once within the river routing scheme, ORCHILEAK assumes that the lateral
transfer of $CO_2$ and DOC are proportional to the volume of water. DOC is divided into a
refractory and labile pool within the river, with half-lives of 80 and 2 days respectively. The
refractory pool corresponds to the combined slow and passive DOC pools of the soil C scheme,
and the labile pool corresponds to the active soil pool (see section 2.4.1). The concentration of
dissolved $CO_2$ and the temperature-dependent solubility of $CO_2$ are used to calculate the partial
pressure of $CO_2$ ($pCO_2$) in the water column. In turn, $CO_2$ evasion is calculated based on $pCO_2$,
along with a diurnally variable water surface area and a gas exchange velocity. Fixed gas
exchange velocities of 3.5 m $d^{-1}$ and 0.65 m $d^{-1}$ respectively are used for rivers (including open
floodplains) and forested floodplains.
In this study, as in previous studies (Lauerwald et al., 2017, Hastie et al. 2019, Bowring et al.,
2019), we run the model at a spatial resolution of 1° and use the default time step of 30 min for
all vertical transfers of water, energy and C between vegetation, soil and the atmosphere, and
the daily time-step for the lateral routing of water. Until now, in the Tropics, ORCHILEAK



has been parameterized and calibrated only for the Amazon River basin (Lauerwald et al., 2017,
Hastie et al. 2019). To adapt and apply ORCHILEAK to the specific characteristics of the
Congo River basin (2.1), we had to establish new forcing files representing the maximal
fraction of floodplains (MFF) and the maximal fraction of swamps (MFS) (2.2) and to
recalibrate the river routing module of ORCHILEAK (2.3). All of the processes represented in
ORCHILEAK remain identical to those previously represented for the Amazon ORCHILEAK
(Lauerwald et al., 2017; Hastie et al., 2019). In the following methodology sections, we
describe; 2.1- Congo basin description, 2.2- Development of floodplains and swamps forcing
files, 2.3- Calibration of hydrology, 2.4- Simulation set-up, 2.5- Evaluation and analysis of
simulated fluvial C fluxes, and 2.6- Calculating the net carbon balance of the Congo Basin. For
a full description of the ORCHILEAK model please see Lauerwald et al. (2017).
**2.1 Congo basin description**
The Congo Basin is the world's second largest area of contiguous tropical rainforest and second
largest river basin in the world (Fig. 1), covering an area of $3.7 \times 10^6$ km$^2$, with a mean discharge
of around 42,000 m$^{-3}$ s$^{-1}$ (O'Loughlin et al., 2013) and a variation between 24,700–75,500 m$^{-3}$
s$^{-1}$ across months (Coynel et al., 2005).




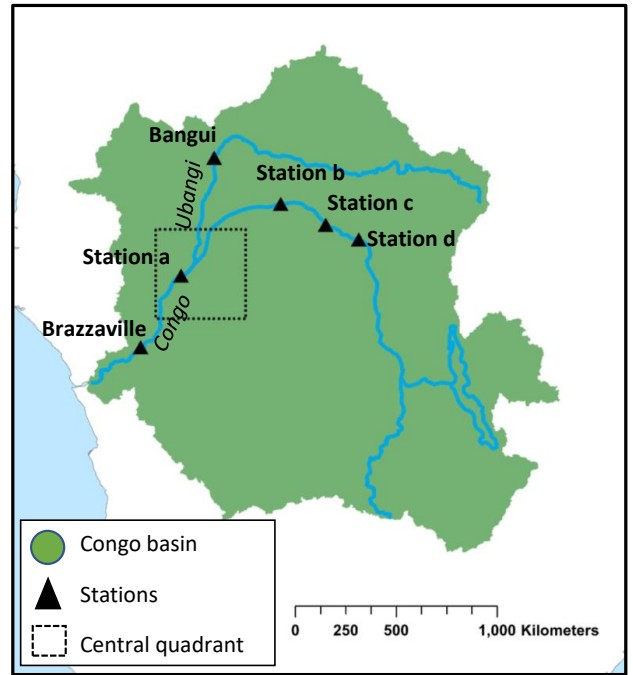

**Figure 1:Extent of the Congo Basin, central quadrant of the "Cuvette Centrale" and sampling stations (for DOC and discharge) along the Congo and Ubangi Rivers (in italic).**

The major climate (ISMSIP2b, Frieler et al., 2017; Lang et al., 2017) and land-cover (LUH-CMIP5) characteristics of the Congo Basin for the present day (1981-2010) are shown in Figure 2. The mean annual temperature is 25.2 °C but with considerable spatial variation from a low of 18.4°C to a high of 27.2°C (Fig. 2 a), while mean annual rainfall is 1520mm, varying from 733 mm to 4087 mm (Fig. 2 b). ORCHILEAK prescribes 13 different plant functional types (PFTs). Land-use is mixed with tropical broad-leaved evergreen (PFT2, Fig. 1 c), tropical broad-leaved rain green (PFT3, Fig. 1 d), $C_3$ grass (PFT10, Fig. 2 e) and $C_4$ grass (PFT11, Fig. 2 f) covering a maximum of 26%, 35%, 8% and 25% of the basin area respectively (Table A3). Agriculture covers only a small proportion of the basin according to the LUH dataset that is based on FAO cropland area statistics, with C3 (PFT12, Fig. 2 g) and $C_4$ (PFT13, Fig. 2 h) agriculture making up a maximum basin area of 0.5 and 2% respectively (Table A3). In reality, a larger fraction of the basin is composed of small scale and rotational agriculture (Tyukavina



et al., 2018). The ORCHILEAK model also has a "poor soils" forcing file (Fig. 2 j) which
prescribes reduced decomposition rates in soils with low nutrient and pH soils such as Podzols
and Arenosols (Lauerwald et al., 2017). This file is developed from the Harmonized World
Soil Database (FAO/IIASA/ISRIC/ISS-CAS/JRC, 2009).

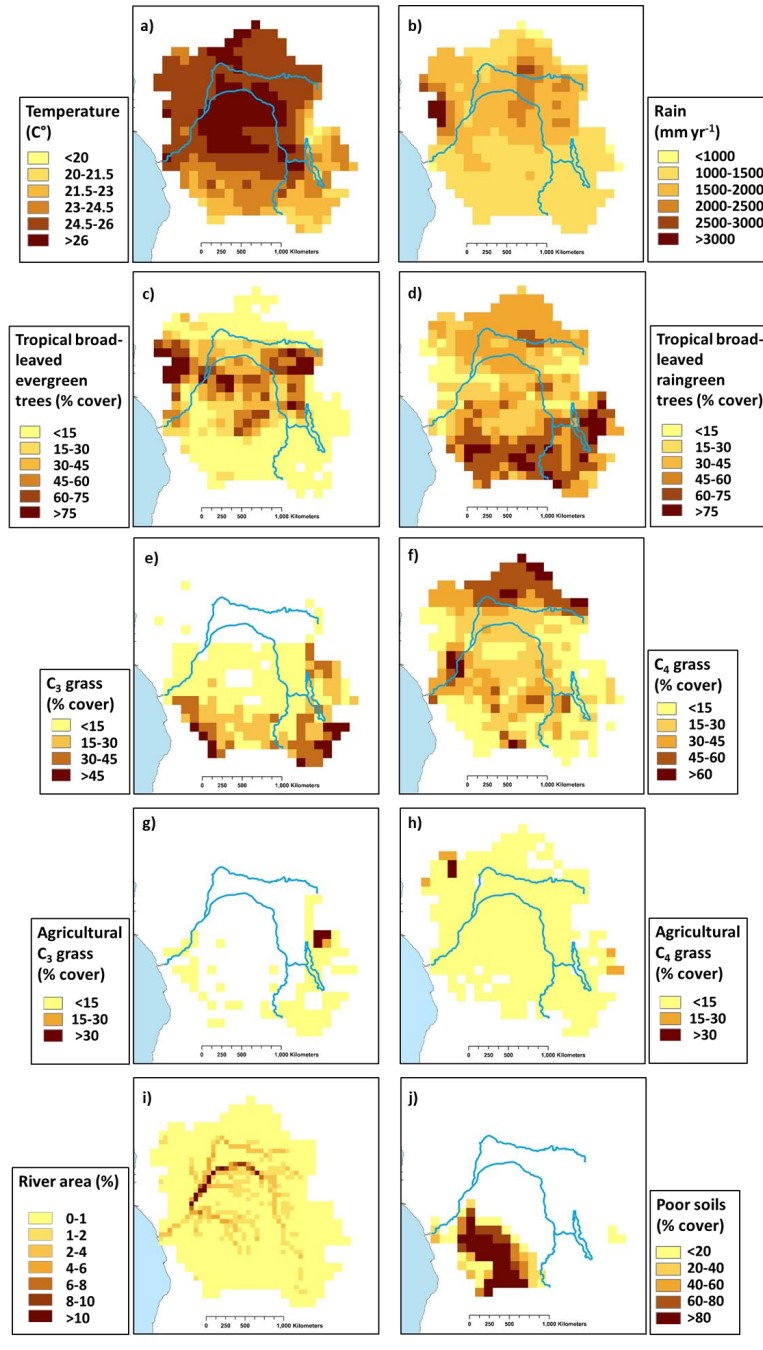


**Figure 2: Present day (1981-2010) spatial distribution of the principal climate and land-use drivers used in ORCHILEAK, across the Congo Basin; a) mean annual temperature in °C, b) mean annual rainfall in mm yr⁻¹, c)-h) mean annual maximum vegetated fraction for PFTs 2,3,**



**10,11,12 and 13, i) river area, and j) Poor soils. All at a resolution of 1° except for river area**
**(0.5°).**
**2.2 Development of floodplains and swamps forcing files**
In ORCHILEAK, water in the river network can be diverted to two types of wetlands,
floodplains and swamps. In each grid where a floodplain exists, a temporary waterbody can be
formed adjacent to the river and is fed by the river once bank-full discharge (see section 2.3)
is exceeded. In grids where swamps exist, a constant proportion of river discharge is fed into
the base of the soil column. The maximal proportions of each grid which can be covered by
floodplains and swamps are prescribed by the maximal fraction of floodplains (MFF) and the
maximal fraction of swamps (MFS) forcing files respectively (Guimberteau et al., 2012). See
also Lauerwald et al. (2017) and Hastie et al. (2019) for further details. We created an MFF
forcing file for the Congo basin, derived from the Global Wetlands[v3] database; the 232 m
resolution tropical wetland map of Gumbricht et al. (2017) (Fig. 3 a and b). We firstly
amalgamated all the categories of wetland before aggregating them to a resolution of 0.5° (the
resolution at which the floodplain/swamp forcing files are read by ORCHILEAK), assuming
that this represents the maximum extent of inundation in the basin. This results in a mean MFF
of 10.3%, i.e. a maximum of 10.3% of the surface area of the Congo basin can be inundated
with water. This is very similar to the mean MFF value of 10% produced with the Global Lakes
and Wetlands Database, GLWD (Lehner, & Döll, P.,2004; Borges et al., 2015[b]). We also
created an MFS forcing file from the same dataset (Fig. 3 c and d), merging the 'swamps' and
'fens' wetland categories from Global Wetlands[v3] database (Gumbricht et al., 2017) and again
aggregating them to a 0.5° resolution.






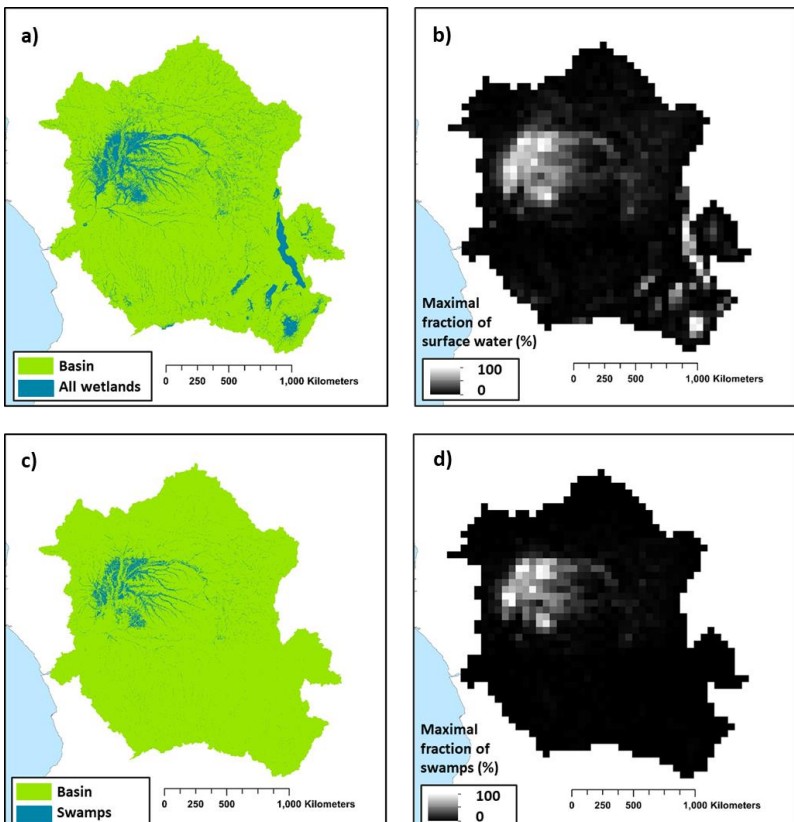

**Figure 3: a) Wetland extent (from Gumbricht et al., 2017). b) The new maximal fraction of floodplain (MFF) forcing file developed from a). c) Swamps (including fens) category within Congo basin from Gumbricht et al (2017). d) the new maximal fraction of swamps (MFS) forcing file developed from c). Panels a) and b) are at the same resolution as the Gumbricht dataset (232m) while b) and d) are at a resolution of 0.5°. Note that 0.5° is the resolution of the sub unit basins in ORCHILEAK (Lauerwald et al., 2015), with each 1° grid containing four sub basins.**


## 2.3 Calibration of hydrology

As the main driver of the export of C from the terrestrial to aquatic system, it is crucial that the
model can represent present-day hydrological dynamics, at the very least on the main stem of
the Congo. As this study is primarily concerned with decadal- centennial timescales our priority
was to ensure that the model can accurately recreate observed mean annual discharge at the
most downstream gauging station Brazzaville. We also tested the model's ability to simulate



observed discharge seasonality, as well as flood dynamics. Moreover, no data is available with
which to directly evaluate the simulation of DOC and $CO_2$ leaching from the soil to the river
network, and thus we tested the model's ability to recreate the spatial variation of observed
riverine DOC concentrations at specific stations where measurements are available (Borges at
al., 2015[b] and shown in Fig. 1), river DOC concentration being regarded as an integrator of the
C transport at the terrestrial-aquatic interface.
We first ran the model for the present-day period, defined as from 1990 to 2005/2010
depending on which climate forcing data was applied, using four climate forcing datasets;
namely ISIMIP2b (Frieler et al., 2017), Princeton GPCC (Sheffield et al., 2006), GSWP3 (Kim,
2017) and CRUNCEP (Viovy, 2018). We used ISIMIP2b for the historical and future
simulations as it is the only climate forcing dataset to cover the full period (1861-2099).
However, we compared it to other climate forcing datasets for the present day in order to gauge
its ability to simulate observed discharge on the Congo River at Brazzaville (Table A1).
Without calibration, the majority of the different climate forcing model runs performed poorly,
unable to accurately represent the seasonality and mean monthly discharge at Brazzaville
(Table A1). The best performing climate forcing dataset was ISIMIP2b followed by Princeton
GPCC with root mean square errors (RMSE) of 29% and 40% and Nash Sutcliffe efficiencies
(NSE) of 0.20 and -0.25, respectively. NSE is a statistical coefficient specifically used to test
the predictive skill of hydrological models (Nash & Sutcliffe, 1970).
For ISIMIP2b we further calibrated key hydrological model parameters, namely the constants
which dictate the water residence time of the groundwater (=slow reservoir), headwaters (=
fast reservoir) and floodplain reservoirs in order to improve the simulation of observed
discharge at Brazzaville (Table 2). To do so, we tested different combinations of water
residence times for the three reservoirs, eventually settling on 1, 0.5 and 0.5 (days) for the slow,





fast and floodplain reservoirs respectively, all three being reduced compared to those values
used in the original ORCHILEAK calibration for the Amazon (Lauerwald at al., 2017).
In order to calibrate the simulated discharge against observations, we first modified the flood
dynamics of ORCHILEAK in the Congo Basin for the present day by adjusting bank-full
discharge (streamr$_{50th}$, Lauerwald et al., 2017) and 95$^{th}$ percentile of water level heights
(floodh$_{95th}$). As in previous studies on the Amazon basin (Lauerwald et al. 2017, Hastie et al.,
2019) we defined bank-full discharge, i.e. the threshold discharge at which floodplain
inundation starts, as the median discharge (50$^{th}$ percentile i.e. streamr$_{50th}$) of the present-day
climate forcing period (1990 to 2005). After re-running each model parametrization (different
water residence times) to obtain those bank-full discharge values, we calculated floodh$_{95th}$ over
the simulation period for each grid cell (Table 1). This value is assumed to represent the water
level over the river banks at which the maximum horizontal extent of floodplain inundation is
reached. We then ran the model for a final time and validated the outputs against discharge
data at Brazzaville (Cochonneau et al., 2006, Fig. 1). This procedure was repeated iteratively
with the ISIMIP2b climate forcing, modifying the water residence times of each reservoir in
order to find the best performing parametrization.
Limited observed discharge data is available for the Congo basin, with the majority
concentrated on the main stem of the Congo, at Brazzaville station. After comparing simulated
vs observed discharge at Brazzaville (NSE, RMSE, Table 2), we used the data of Bouillon et
al. (2014) to further validate discharge at Bangui (Fig. 1) on the main tributary Ubangi. In
addition, we compared the simulated seasonality of flooded area against the satellite derived
dataset GIEMS (Prigent et al., 2007; Becker et al., 2018), within the Cuvette Centrale wetlands
(Fig. 1).



### 2.4 Simulation set-up

A list of the main forcing files used, along with data sources, is presented in Table 1. The
derivation of the floodplains and swamp (MFF & MFS) is described in section 2.2 while the
calculation of "bankfull discharge" (streamr$_{50th}$) and "95th percentile of water table height over
flood plain" (floodh$_{95th}$) (Table 1) is described in section 2.3.

#### 2.4.1   Soil carbon spin up

ORCHILEAK includes a soil module, primarily derived from ORCHIDEE-SOM (Camino
Serrano, 2018). The soil module has 3 different pools of soil DOC; the passive, slow and active
pool and these are defined by their source material and residence times ($\tau_{carbon}$). ORCHILEAK
also differentiates between flooded and non-flooded soils; decomposition rates of DOC, SOC
and litter being reduced (3 times lower) in flooded soils. In order for the soil C pools to reach
steady state, we spun-up the model for around 9,000 years, with fixed land-use representative
of 1861, and looping over the first 30 years of the ISMSIP2b climate forcing data (1861-1890).
During the first 2,000 years of spin-up, we ran the model with an atmospheric $CO_2$
concentration of 350 µatm and default soil C residence times ($\tau_{carbon}$) halved, which allowed it
to approach steady-state more rapidly. Following this, we ran the model for a further 7,000
years reverting to the default $\tau_{carbon}$ values. At the end of this process, the soil C pools had
reached approximately steady state; <0.02% change in each pool over the final century of the
spin-up.

#### 2.4.2 Transient simulations

After the spin-up, we ran a historical simulation from 1861 until the present day, 2005 in the
case of the ISIMIP2b climate forcing data. We then ran a future simulation until 2099, using
the final year of the historical simulation as a restart file. In both of these simulations, climate,
atmospheric $CO_2$ and land-cover change were prescribed as fully transient forcings according
to the RCP6.0 scenario. For climate variables, we used the IPSL-CM5A-LR model outputs for



RCP 6.0, bias corrected by the ISIMIP2b procedure (Frieler et al., 2017; Lange et al., 2017),
while land-use change was taken from the 5th Coupled Model Intercomparison Project
(CMIP5). As our aim is to investigate long-term trends, we calculated 30-years running means
of simulated C flux outputs in order to smooth interannual variations. RCP 6.0 is an emissions
pathway that leads to a "stabilization of radiative forcing at 6.0 Watts per square meter ($Wm^{-2}$)
in the year 2100 without exceeding that value in prior years" (Masui et al., 2011). It is
characterised by intermediate energy intensity, substantial population growth, mid-high C
emissions, increasing cropland area to 2100 and decreasing natural grassland area (van Vuuren
et al., 2011). In the paper which describes the development of the future land use change
scenarios under RCP 6.0 (Hurtt et al., 2011), it is shown that land use change is highly sensitive
to land use model assumptions, such as whether or not shifting cultivation is included. In our
simulations, shifting cultivation is not included. Moreover, Africa is one the regions with the
largest uncertainty range, and thus, there is considerable uncertainty associated with the effect
of future land-use change (Hurtt et al., 2011). We chose RCP 6.0 as it represents a no mitigation
(mid-high emissions) scenario and because it was the scenario applied in the recent paper of
Lauerwald et al. (submitted) to examine the long-term LOAC fluxes in the Amazon basin.
Therefore, we can directly compare our results for the Congo to those for the Amazon.
Moreover, the ISIMIP2b data only provided two RCPs at the time we performed the
simulations; RCP 2.6 (low emission) and RCP 6.0.
With the purpose of evaluating separately the effects of land-use change, climate change, and
rising atmospheric $CO_2$, we ran a series of factorial simulations. In each simulation, one of
these factors was fixed at its 1861 level (the first year of the simulation), or in the case of fixed
climate change, we looped over the years 1861-1890. The outputs of these simulations (also
30-year running means) were then subtracted from the outputs of the main simulation (original





run with all factors varied) so that we could determine the contribution of each driver (Fig. 10,
Table 1).

| Table 1:Main forcing files used for simulations | | | |
|---|---|---|---|
| **Variable** | **Spatial resolution** | **Temporal resolution** | **Data source** |
| Rainfall, snowfall, incoming shortwave and longwave radiation, air temperature, relative humidity and air pressure (close to surface), wind speed (10 m above surface) | 1° | 1 day | ISIMIP2b, IPSL-CM5A-LR model outputs for RCP6.0 (Frieler et al., 2017) |
| Land cover (and change) | 0.5° | annual | LUH-CMIP5 |
| Poor soils | 0.5° | annual | Derived from HWSD v 1.1 (FAO/IIASA/ISRIC/ISS-CAS/JRC, 2009) |
| Stream flow directions | 0.5° | annual | STN-30p (Vörösmarty et al., 2000) |
| Floodplains and swamps fraction in each grid (MFF & MFS) | 0.5° | annual | derived from the wetland high resolution data of Gumbricht et al. (2017) |
| River surface areas | 0.5° | annual | Lauerwald et al. (2015) |
| Bankfull discharge (streamr$_{50th}$) | 1° | annual | derived from calibration with ORCHILEAK (see section 2,3) |
| 95th percentile of water table height over flood plain (floodh$_{95th}$) | 1° | annual | derived from calibration with ORCHILEAK (see section 2.3) |

**2.5 Evaluation and analysis of simulated fluvial C fluxes**
We first evaluated DOC concentrations at several locations along the Congo mainstem (Fig.
1), and on the Ubangi river against the data of Borges at al. (2015[b]). We also compared the
various simulated components of the net C balance (e.g. NPP) of the Congo against values
described in the literature (Williams et al., 2007; Lewis et al., 2009; Verhegghen et al., 2012;
Valentini et al., 2014; Yin et al., 2017). In addition, we assessed the relationship between the
interannual variation in present day (1981-2010) C fluxes of the Congo basin and variation in
temperature and rainfall. This was done through linear regression using STATISTICA[TM]. We
found trends in several of the fluxes over the 30-year period (1981-2010) and thus detrended
the time series with the "Detrend" function, part of the "SpecsVerification" package in R (R
Core Team 2013), before undertaking the statistical analysis focused on the climate drivers of
inter-annual variability.





**2.6 Calculating the net carbon balance of the Congo basin**
We calculated Net Ecosystem Production (NEP) by summing the terrestrial and aquatic C
fluxes of the Congo basin (Eq. 1), while we incorporated disturbance fluxes (Land-use change
flux and harvest flux) to calculate Net Biome Production (NBP) (Eq. 2). Positive values of
NBP and NEP equate to a net terrestrial C sink.
NEP is defined as follows:
$$NEP = NPP + TF - SHR - FCO_2 - LE_{\text{Aquatic}} \qquad (1)$$
Where *NPP* is terrestrial net primary production, *TF* is the throughfall flux of DOC from the
canopy to the ground, *SHR* is soil heterotrophic respiration (only that evading from the *terra-*
*firme* soil surface); $FCO_2$ is $CO_2$ evasion from the water surface and $LE_{Aquatic}$ is the lateral
export flux of C (DOC + dissolved $CO_2$) to the coast. NBP is equal to NEP except with the
inclusion of the C lost (or possibly gained) via land use change (*LUC*) and crop harvest (*HAR*).
Wood harvest is not included for logging and forestry practices, but during deforestation LUC,
a fraction of the forest biomass is harvested and channelled to wood product pools with
different decay constants. *LUC* includes land conversion fluxes and the lateral export of wood
products biomass, that is, assuming that wood products from deforestation are not consumed
and released as $CO_2$ over the Congo, but in other regions:
$$NBP = NEP - (LUC + HAR) \qquad (2)$$

**3. Results**
**3.1 Simulation of Hydrology**
The final model configuration is able to closely reproduce the mean monthly discharge at
Brazzaville (Fig. 4 a), Table 2) and captures the seasonality moderately well (Fig. 4 a, Table 2,

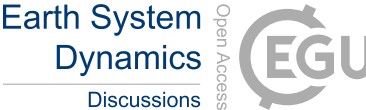

RMSE =23%, $R^2$ =0.84 *versus* RMSE= 29% and $R^2$ =0.23 without calibration, Table A1). At
Bangui on the Ubangi River (Fig. 1), the model is able to closely recreate observed seasonality
(Fig. 4 b), RMSE =59%, $R^2$ =0.88) but substantially underestimates the mean monthly
discharge, our value being only 50% of the observed. We produce reasonable NSE values of
0.66 and 0.31 for Brazzaville and Bangui respectively, indicating that the model is moderately
accurate in its simulation of seasonality.
We also evaluated the simulated seasonal change in flooded area in the central (approx.
200,000 $km^2$, Fig. 1) part of the Cuvette Centrale wetlands against the GIEMS inundation
dataset (1993-2007, maximum inundation minus minimum or permanent water bodies, Prigent
et al., 2007; Becker et al., 2018). While our model is able to represent the seasonality in flooded
area relatively well ($R^2$ =0.75 Fig. 4 c), it considerably overestimates the magnitude of flooded
area relative to GIEMS (Fig. 4 c, Table 2). However, the dataset that we used to define the
MFF and MFS forcing files (Gumbricht et al., 2017) is produced at a higher resolution than
GIEMS and will capture smaller wetlands than the GIEMS dataset, and thus the greater flooded
area is to be expected. GIEMS is also known to underestimate inundation under vegetated areas
(Prigent et al., 2007, Papa et al., 2010) and has difficulties to capture small inundated areas
(Prigent et al., 2007; Lauerwald et al., 2017). Indeed, with the GIEMS data we produce an
overall flooded area for the Congo Basin of just 3%, less than one-third of that produced with
the Gumbricht dataset (Gumbricht et al., 2017) or the GLWD (Lehner, & Döll, P.,2004). As
such, it is to be expected that there is a large RMSE (272%, Table 2) between simulated flooded
area and GIEMS; more importantly, the seasonality of the two is highly correlated ($R^2$ = 0.67,
Table 2). Overall, the hydrological performance of the model against those datasets is
satisfactory as the main purpose of this study is to estimate the long-term changes of aquatic C
fluxes. In particular, it can closely recreate the mean monthly/annual discharge at Brazzaville
(Table 2), the most downstream gauging station on the Congo (Fig. 1). As such, we consider
the hydrological performance to be sufficiently good for our aims.

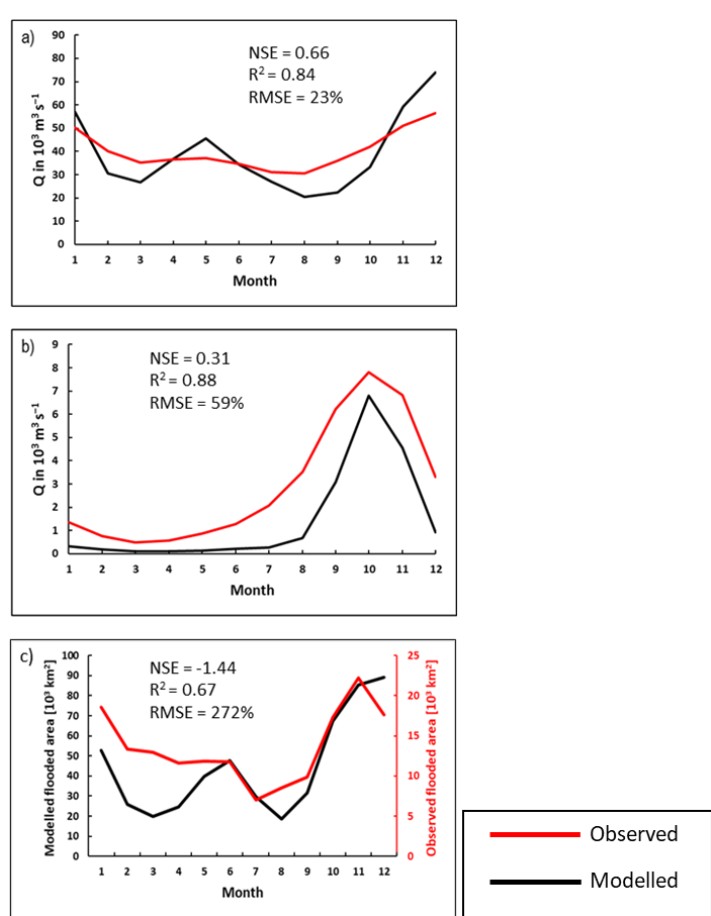

**Figure 4: Seasonality of simulated versus observed discharge at a) Brazzaville on the Congo (Cochonneau et al., 2006), b) Bangui on the Ubangi (Bouillon et al., 2014) 1990-2005 monthly mean and c) flooded area in the the central (approx. 200,000 km² ) area of the Cuvette Centrale wetlands versus GIEMS (1993-2007, Becker et al., 2018). The observed flooded area data represents the maximum minus minimum (permanent water bodies such as rivers) GIEMS inundation. See Figure 1 for locations**









| Table 2: Performance statistics for modelled versus observed seasonality of discharge and flooded area in Cuvette Centrale | | | | | |
|---|---|---|---|---|---|
| **Station** | RSME | NSE | $R^2$ | Simulated mean monthly discharge ($m^3\ s^{-1}$) | Observed mean monthly discharge ($m^3\ s^{-1}$) |
| Brazzaville | 23% | 0.66 | 0.84 | 38,944 | 40,080 |
| Bangui | 59% | 0.31 | 0.88 | 1,448 | 2,923 |
| | | | | Simulated mean monthly flooded area ($10^3\ km^2$) | Observed mean monthly flooded area ($10^3\ km^2$) |
| Flooded area (Cuvette Centrale) | 272% | -1.44 | 0.67 | 44 | 14 |


## 3.2 Carbon fluxes along the Congo basin for the present day

For the present day (1981-2010) we estimate a mean annual terrestrial net primary production (NPP) of 5,800 ±166 (standard deviation, SD) Tg C yr$^{-1}$ (Fig. 5), corresponding to a mean areal C fixation rate of approximately 1,500 g C m$^{-2}$ yr$^{-1}$ (Fig. 6 a). We find a significant positive correlation between the interannual variation of NPP and rainfall (detrended $R^2$= 0.41, p<0.001, Table A2) and a negative correlation between annual NPP and temperature (detrended $R^2$= 0.32, p<0.01, Table A2). We also see considerable spatial variation in NPP across the Congo Basin (Fig.6 a).

We simulate a mean soil heterotrophic respiration (SHR) of 5,300 ±99 Tg C yr$^{-1}$ across the Congo basin (Fig. 5). Contrary to NPP, interannual variation in annual SHR is positively correlated with temperature (detrended $R^2$= 0.57, p<0.0001, Table A2) and inversely correlated with rainfall (detrended $R^2$= 0.10), though the latter relationship is not significant (p>0.05). We estimate a mean annual aquatic $CO_2$ evasion of rate of 1,363 ±83 g C m$^{-2}$ yr$^{-1}$, amounting



to a total of 235±54 Tg C yr$^{-1}$ across the total water surfaces of the Congo basin (Fig. 5) and
attribute 85% of this flux to flooded areas, meaning that only 32 Tg C yr$^{-1}$ is evaded directly
from the river surface. Interannual variation in aquatic $CO_2$ evasion (1981-2010) shows a
strong positive correlation with rainfall (detrended $R^2$= 0.75, p<0.0001, Table A2) and a weak
negative correlation with temperature (detrended $R^2$=0.09, not significant, p>0.05). Aquatic
$CO_2$ evasion also exhibits substantial spatial variation (Fig.6, d), displaying a similar pattern to
both terrestrial DOC leaching ($DOC_{inp}$) ($R^2$= 0.81, p<0.0001, Fig.6, b) as well as terrestrial
$CO_2$ leaching ($CO_{2inp}$) ($R^2$= 0.96, p<0.0001, Fig.6, c) into the aquatic system, but not terrestrial
NPP ($R^2$= 0.01, p<0.05, Fig.6, a). We simulate a flux of DOC throughfall from the canopy of
27 ±1 Tg C yr$^{-1}$.
We estimate a mean annual C (DOC + dissolved $CO_2$) export flux to the coast of 15 ±4 Tg C
yr$^{-1}$ (Fig. 5). In Figure 7, we compare simulated DOC concentrations at six locations (Fig. 1)
along the Congo River and Ubangi tributary, against the observations of Borges at al. (2015[b]).
We show that we can recreate the spatial variation in DOC concentration within the Congo
basin relatively closely with an $R^2$ of 0.82 and an RMSE of 19% (Fig. 7).
For the present day (1981-2010) we estimate a mean annual net ecosystem production (NEP)
of 277 ±137 Tg C yr$^{-1}$ and a net biome production (NBP) of 107 ±133 Tg C yr$^{-1}$ (Fig. 5).
Interannually, both NEP and NBP exhibit a strong inverse correlation with temperature
(detrended NEP $R^2$=0.55, p<0.0001, detrended NBP $R^2$=0.54, p<0.0001) and weak positive
relationship with rainfall (detrended NEP $R^2$=0.16, p<0.05, detrended NBP $R^2$=0.14, p<0.05).
Furthermore, we simulate a present day (1981-2010) living biomass of 41 ±1 Pg C and a total
soil C stock of 109 ±1 Pg C.





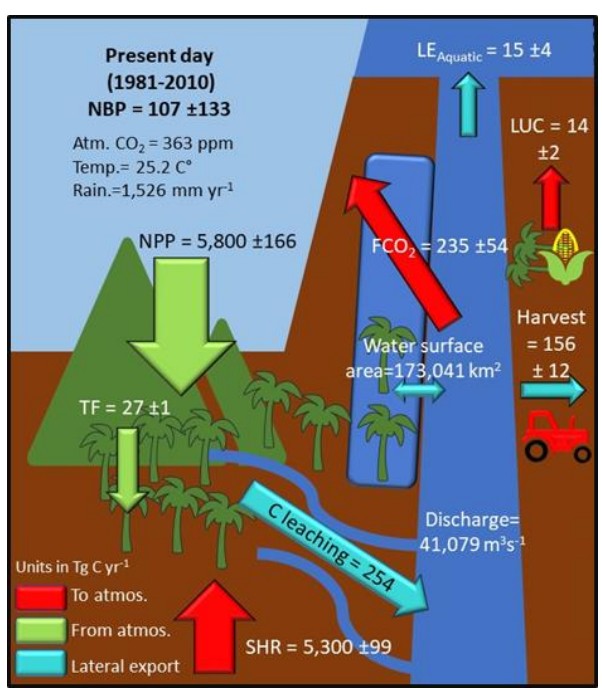

**Figure 5: Annual C budget (NBP) for the Congo basin for the present day (1981-2010) simulated with ORCHILEAK, where NPP is terrestrial net primary productivity, TF is throughfall, SHR is soil heterotrophic respiration, FCO$_2$ is aquatic CO$_2$ evasion, LOAC is C leakage to the land-ocean aquatic continuum (FCO$_2$ + $LE_{Aquatic}$), LUC is flux from Land-use change, and $LE_{Aquatic}$ is the export C flux to the coast. Range represents the standard deviation (SD).**





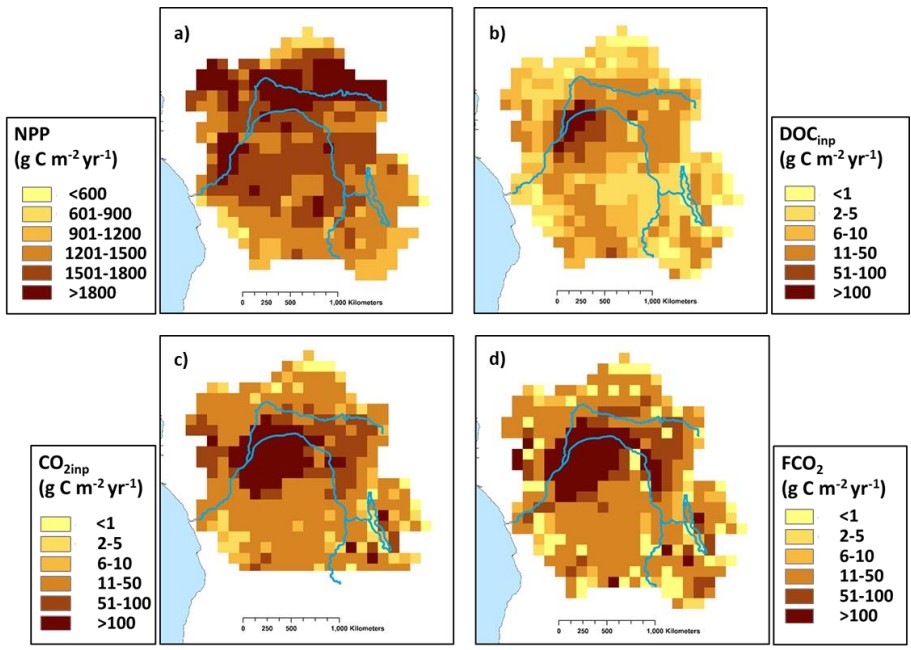


**Figure 6:Present day (1981-2010) spatial distribution of a) terrestrial net primary productivity (NPP), b) dissolved organic carbon leaching from soils into the aquatic system (DOC$_{inp}$), c) CO$_2$ leaching from soils into the aquatic system (CO$_{2inp}$) and d) aquatic CO$_2$ evasion (FCO$_2$). Main rivers in blue. All at a resolution of 1°**

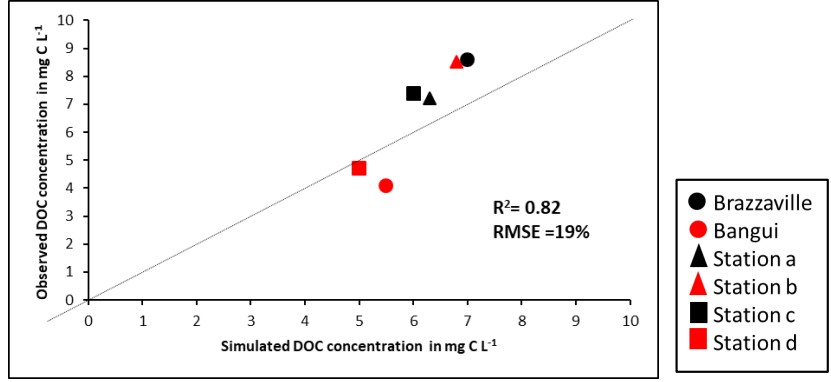

**Figure 7: Observed (Borges et al., 2015[a]) versus simulated DOC concentrations at several sites along the Congo and Ubangi rivers. See Fig. 1 for locations. The simulated DOC concentrations represent the mean values across the particular sampling period at each site detailed in Borges et al. (2015[a]).**






### 3.3 Long-term temporal trends in carbon fluxes


We find an increasing trend in aquatic $CO_2$ evasion (Fig. 8 a) throughout the simulation period,
rising slowly at first until the 1960s when the rate of increase accelerates. In total $CO_2$ evasion
rose by 79% from 186 Tg C $yr^{-1}$ at the start of the simulation (1861-1890 mean) (Fig. 9) to 333
Tg C $yr^{-1}$ at the end of this century (2070-2099 mean, Fig. 9), while the increase until the
present day (1981-2010 mean) is of +26 % (to 235 Tg C $yr^{-1}$), though these trends are not
uniform across the basin (Fig A1). The lateral export flux of C to the coast ($LE_{Aquatic}$) follows
a similar relative change (Fig. 8b), rising by 67% in total, from 12 Tg C $yr^{-1}$ (Fig. 9) to 15 Tg
C $yr^{-1}$ for the present day, and finally to 20 Tg C $yr^{-1}$ (2070-2099 mean, Fig. 9). This is greater
than the equivalent increase in DOC concentration (24%, Fig. 8b) due to the concurrent rise in
rainfall (by 14%, Fig 8h) and in turn discharge (by 29%, Fig. 8h).
Terrestrial NPP and SHR also exhibit substantial increases of 35% and 26% respectively across
the simulation period and similarly rise rapidly after 1960 (Fig. 8 c). NEP, NBP (Fig. 8 d) and
living biomass (Fig. 8 e) follow roughly the same trend as NPP, but NEP and NBP begin to
slow down or even level-off around 2030 and in the case of NBP, we actually simulate a
decreasing trend over approximately the final 50 years. Interestingly, the proportion of NPP
lost to the LOAC also increases from approximately 3% to 5% (Fig. 8c). We also find that
living biomass stock increases by a total of 53% from 1861 to 2099. Total soil C also increases
over the simulation but only by 3% from 107 to 110 Pg C $yr^{-1}$ (Fig. 8 e). Emissions from land-
use change (LUC) show considerable decadal fluctuation increasing rapidly in the second half
of the 20th century and decreasing in the mid-21st century before rising again towards the end
of the simulation (Fig. 8 f). The harvest flux (Fig.8 f) rises throughout the simulation with the
exception of a period in the mid-21st century during which it stalls for several decades. This is
reflected in the change in land-use areas from 1861- 2099 (Fig. A2, Table A3) during which



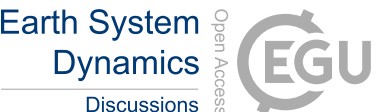

the natural forest and grassland PFTs marginally decrease while both C$_3$ and C$_4$ agricultural
grassland PFTs increase.

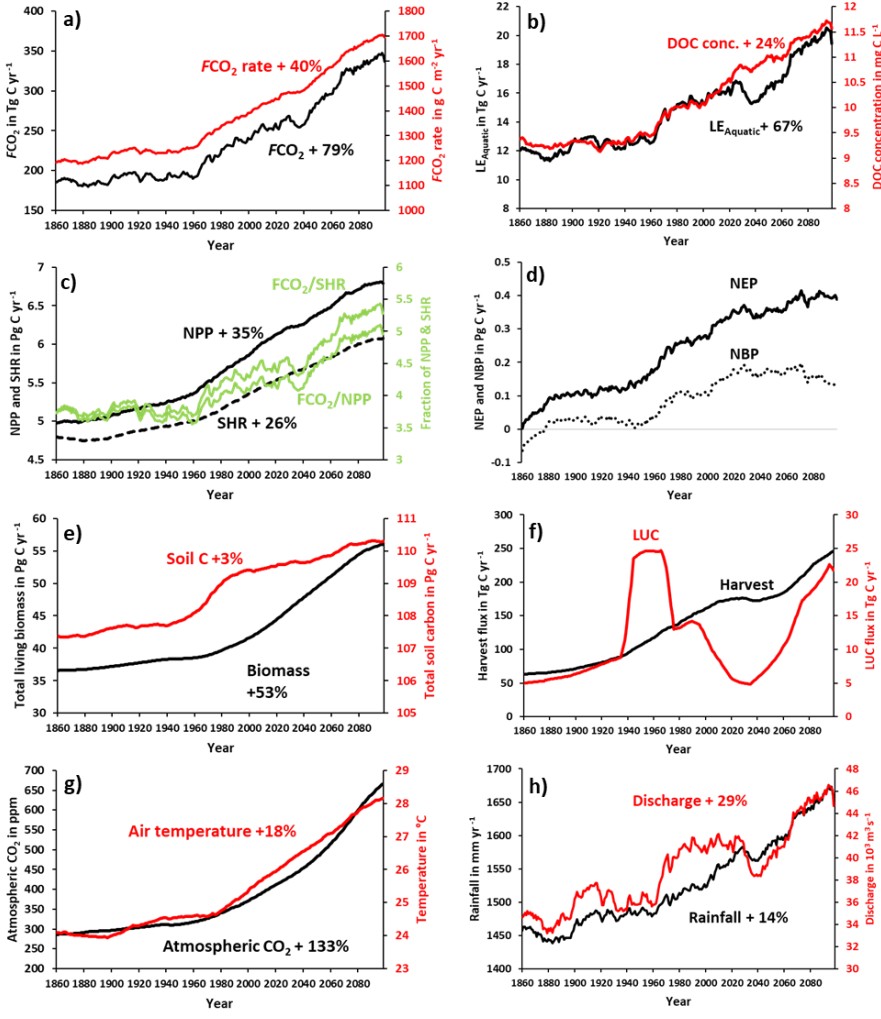


**Figure 8: Simulation results for various C fluxes and stocks from 1861-2099, using IPSL-CM5A-LR model outputs for RCP 6.0 (Frieler et al., 2017).  All panels except for atmospheric CO$_2$, biomass and soil C correspond to 30-year running means of simulation outputs. This was done in order to suppress interannual variation, as we are interested in longer-term trends.**




### 3.4 Drivers of simulated trends in carbon fluxes

The dramatic increase in the concentration of atmospheric $CO_2$ (Fig. 8 g) and subsequent fertilization effect on terrestrial NPP has the greatest overall impact on all of the fluxes across the simulation period (Fig. 10). It is responsible for the vast majority of the growth in NPP, SHR, aquatic $CO_2$ evasion and flux of C to the coast (Fig. 10 a, b, c & d). The effect of LUC on these four fluxes is more or less neutral, while the impact of climate change is more varied. The aquatic fluxes (Fig. 10 c, d) respond positively to an acceleration in the increase of both rainfall (and in turn discharge, Fig. 8 h) and temperature (Fig. 8 g) starting around 1970. From around 2020, the impact of climate change on the lateral flux of C to the coast (Fig 10 d) reverts to being effectively neutral, likely a response to a slowdown in the rise of rainfall and indeed a decrease in discharge (Fig 8 h), as well as perhaps the effect of temperature crossing a threshold. The response of the overall loss of terrestrial C to the LOAC (i.e. the ratio of LOAC/NPP, Fig. 10 e) is relatively similar to the response of the individual aquatic fluxes but crucially, climate change exerts a much greater impact, contributing substantially to an increase in the loss of terrestrial NPP to the LOAC in the 1960s, and again in the second half of the 21$^{st}$ century. These changes closely coincide with the pattern of rainfall and in particular with changes in discharge (Fig. 8 h).

Overall temperature and rainfall increase by 18% and 14% from 24°C to 28°C and 1457mm to 1654mm respectively, but in Fig. A2 one can see that this increase is non-uniform across the basin. Generally speaking, the greatest increase in temperature occurs in the south of the basin while it is the east that sees the largest rise in rainfall (Fig. A2). Land-use changes are similarly non-uniform (Fig. A2).

The response of NBP and in NEP (Fig.10 f, g) to anthropogenic drivers is more complex. The simulated decrease in NBP towards the end of the run is influenced by a variety of factors; LUC and climate begin to have a negative effect on NBP (contributing to a decrease in NBP)





at a similar time while the positive impact (contributing to an increase in NBP) of atmospheric
CO₂ begins to slow down and eventually level-off (Fig.10 g). LUC continues to have a positive
effect on NEP (Fig.10 f) due to the fact that the expanding C₄ crops have a higher NPP than
forests, while it has an overall negative effect on NBP at the end of the simulation due to the
inclusion of emissions from crop harvest.

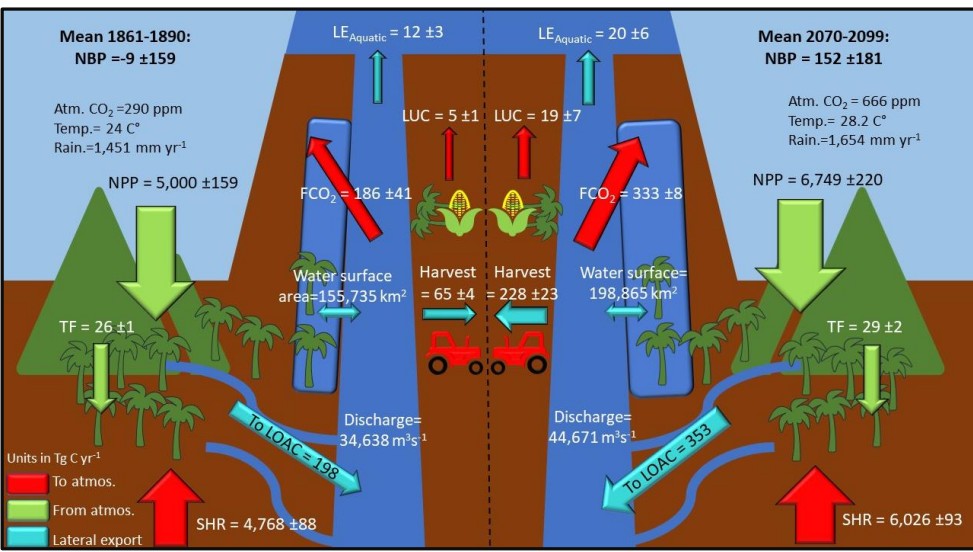


**Figure 9: Annual C budget (NBP) for the Congo basin for; left, the Year 1861 and right, the**
**Year 2099, simulated with ORCHILEAK. NPP is terrestrial net primary productivity, TF is**
**throughfall, SHR is soil heterotrophic respiration, FCO₂ is aquatic CO2 evasion, LOAC is C**
**leakage to the land-ocean aquatic continuum (FCO₂ + LE$_{Aquatic}$), LUC is flux from Land-use**
**change, and LE$_{Aquatic}$ is the export C flux to the coast. Range represents the standard deviation**
**(SD).**









**Figure 10: Contribution of anthropogenic drivers; atmospheric CO₂ concentration (CO₂ atm), climate change (CC) and land use change (LUC) to changes in the various carbon fluxes along the Congo Basin, under IPSL-CM5A-LR model outputs for RCP 6.0 (Frieler et al., 2017).**





## 4. Discussion

### 4.1 Congo basin carbon balance

We simulate a mean present-day terrestrial NPP of approximately 1,500 g C $m^{-2}$ $yr^{-1}$ (Fig. 6),

substantially larger than the MODIS derived value of around 1,000 g C $m^{-2}$ $yr^{-1}$ from Yin et al.

(2017) across central Africa, though it is important to note that satellite derived estimates of

NPP can underestimate the impact of $CO_2$ fertilization, namely its positive effect on

photosynthesis (De Kauwe et al., 2016; Smith et al., 2019). Our stock of the present-day living

biomass of 41.1 Pg C is relatively close to the total Congo vegetation biomass of 49.3 Pg C

estimated by Verhegghen et al. (2012) based on the analysis of MERIS satellite data. Moreover,

our simulated Congo Basin soil C stock of 109 ±1.1 Pg C is consistent with the approximately

120-130 Pg C across Africa between the latitudes 10°S to 10°N in the review of Williams et

al. (2007), between which the Congo represents roughly 70% of the land area. Therefore, their

estimate of soil C stocks across the Congo only would likely be marginally smaller than ours.

It is also important to note that neither estimate of soil C stocks explicitly take into account the

newly discovered peat store of 30 Pg C (Dargie et al., 2017) and therefore both are likely to

represent conservative values. In addition, Williams et al. (2007) estimate the combined fluxes

from conversion to agriculture and cultivation to be around 100 Tg C $yr^{-1}$ in tropical Africa

(largely synonymous with the Congo Basin), which is relatively close to our present day

estimate of harvesting + land-use change flux of 170 Tg C $yr^{-1}$.

Our results suggest that $CO_2$ evasion from the water surfaces of the Congo is sustained by

leaching of dissolved $CO_2$ and DOC with 226 Tg C and 73 Tg C, respectively, from soils to

the aquatic system each year (1980-2010, Fig. 6). Moreover, we find that a disproportionate

amount of this transfer occurs (Fig. 6) within the Cuvette Centrale wetland (Fig. 1, Fig. 6) in

the centre of the basin, in agreement with a recent study by Borges et al. (2019). In our study,

this is due to the large areal proportion of inundated land, facilitating the exchange between



soils and aquatic systems.  Borges et al. (2019) conducted extensive measurements of DOC
and $pCO_2$, amongst other chemical variables, along the Congo mainstem and its tributaries
from Kinshasa in the West of the basin (beside Brazzaville, Fig. 1) through the Cuvettte
Centrale to Kisangani in the East (close to station d in Fig. 1). They found that both DOC and
$pCO_2$ approximately doubled from Kisangani downstream to Kinshasa, and demonstrated that
this variation is overwhelmingly driven by fluvial-wetland connectivity, highlighting the
importance of the vast Cuvette Centrale wetland in the aquatic C budget of the Congo basin.
Our estimate of the integrated present-day aquatic $CO_2$ evasion from the river surface of the
Congo basin (32 Tg C $yr^{-1}$) is the same as that estimated by Raymond et al. (2013) (also 32 Tg
C $yr^{-1}$), downscaled over the same basin area, but smaller than the 59.7 Tg C $yr^{-1}$ calculated by
Lauerwald et al. (2015) and far smaller than that of Borges et al. (2015[a]), 133-177 Tg C $yr^{-1}$ or
Borges et al. (2019), 251±46 Tg C $yr^{-1}$. As previously discussed, we simulate the spatial
variation in DOC concentrations measured by Borges et al. (2015[a, b], Fig. 7) relatively closely
and our mean riverine gas exchange velocity $k$ of 3.5 m $d^{-1}$ is similar to the 2.9 m $d^{-1}$ used by
Borges et al. (2015[a]). It is therefore somewhat surprising that our estimate of riverine $CO_2$
evasion is so different, and likely to be related to ORCHILEAK underestimating dissolved $CO_2$
inputs into the river network. Below we discuss some possible explanations for this discrepancy
related to methodological limitations.
One reason for the difference in riverine $CO_2$ evasion could be that the resolution of
ORCHILEAK (1° for C fluxes) is not sufficient to fully capture the dynamics of the smallest
streams of the Congo Basin which have been shown to have the highest DOC and $CO_2$
concentrations (Borges et al., 2019). However, it is important to note that in our simulations,
the evasion flux from rivers only contributes 15% of total aquatic $CO_2$ evasion, and including
the flux from wetlands/floodplains, we produce a total of 235 Tg C $yr^{-1}$.



Another limitation of the ORCHILEAK model is the lack of representation of aquatic plants.
Borges et al. (2019) used the stable isotope composition of $\delta^{13}$C-DIC to determine the origin
of dissolved $CO_2$ in the Congo River system and found that the values were consistent with the
degradation of organic matter, in particular from $C_4$ plants. Crucially, they further found that
the $\delta^{13}$C-DIC values were unrelated to the contribution of *terra-firme* $C_4$ plants, rather that they
were more consistent with the degradation of aquatic $C_4$ plants, namely macrophytes. This also
concurs with the wider conclusions of a previous paper comparing the Congo and the Amazon
(Borges et al., 2015[b]), which highlighted that aquatic macrophytes are more prevalent in the
Congo river and its tributaries compared to the Amazon where strong water currents limit their
abundance. The ORCHILEAK model does not represent aquatic plants, and the wider LSM
ORCHIDEE does not have an aquatic macrophyte PFT either. This could at least partly explain
our conservative estimate of river $CO_2$ evasion, given that tropical macrophytes have relatively
NPP. Rates as high as 3,500 g C m$^{-2}$ yr$^{-1}$ have been measured on floodplains in the Amazon
(Silva et al., 2009). While this value is higher than the values represented in the Cuvette
Centrale by ORCHILEAK (Figure 6), they are of the same order of magnitude and so this
cannot fully explain the discrepancy compared to the results of Borges et al. (2019). In the
Amazon basin it has been shown that wetlands export approximately half of their gross primary
production (GPP) to the river network compared to upland (*terra-firme*) ecosystems which
only export a few percent (Abril et al. 2013). More importantly, Abril et al. (2013) found that
tropical aquatic macrophytes exported 80% of their GPP compared to just 36% for flooded
forest. Therefore, the lack of a bespoke macrophyte PFT may indeed be one reason for the
discrepancy between our results and those of Borges, but largely due to their particularly high
export efficiency to the river-floodplain network as opposed to differences in NPP. While being
a significant limitation, creating and incorporating a macrophyte PFT would be a substantial
undertaking given that the authors are unaware of any published dataset which has



systematically mapped their distribution and abundance. It is important to that while
ORCHILEAK does not include the export of C from aquatic macrophytes it also neglects their
NPP. Moreover, most aquatic macrophytes described in the literature have short (<1 year) life-
cycles (Mitchel & Rogers., 1985). As such, this model limitation will only have a very limited
net effect on our estimate of the overall annual C balance (NBP, NEP) of the Congo basin, and
indeed the other components of NBP.
Our simulated export of C to the coast of 15 (15.3) Tg C yr$^{-1}$ is virtually identical to the
TOC+DIC export estimated by Borges et al. (2015[a]) of 15.5 Tg C yr$^{-1}$, which is consistent with
the fact that we simulate a similar spatial variation of DOC concentrations (Fig. 7 and Fig. 1
for locations). It is also relatively similar to the 19 Tg C yr$^{-1}$ (DOC + DIC) estimated by
Valentini et al. (2014) in their synthesis of the African carbon budget. Valentini et al. (2014)
used the largely empirical based Global Nutrient Export from WaterSheds (NEWS) model
framework and they point out that Africa was underrepresented in the training data used to
develop the regression relationships which underpin the model, and thus this could explain the
small disagreement.
Our estimate of 4% of NPP per year being transferred to inland waters is substantially lower
than that estimated for the Amazon, where around 12% of NPP is lost to the aquatic system
each year (Hastie et al., 2019). There are a number of differences between the drivers in the
two basins, which could explain this. Mean annual rainfall is 44% greater in the Amazon, and
mean annual discharge is 4 times higher, while a maximum of approximately 14% of the
surface of the Amazon Basin is covered by water compared to 10% of the Congo (Borges et
al., 2015[b]; Hastie et al., 2019). Moreover, upland runoff is the main source of water in the
wetlands of the Congo as opposed to the Amazon where exchanges between the river and
floodplain dominate (Lee et al., 2011; Borges et al., 2015[b]). Indeed, the water levels of wetlands
in the Congo have been shown to be consistently higher than adjacent river levels (Lee et al.,



2011). This also partly explains why for the Congo we find that only 15% of aquatic $CO_2$
evasion comes from the river water surface compared to 25% for the Amazon (Hastie et al.,

582    2019).


**4.2 Trends in terrestrial and aquatic carbon fluxes**
There is sparse observed data available on the long-term trends of terrestrial C fluxes in the
Congo. Yin et al. (2017) used MODIS data to estimate NPP between 2001 and 2013 across
central Africa. They found that NPP increased on average by 10 g C $m^{-2}$ per year, while we
simulate an average annual increase of 4 g C $m^{-2}$ $yr^{-1}$ over the same period across the Congo
Basin. The two values are not directly comparable as they do not cover precisely the same
geographic area but it is encouraging that our simulations exhibit a similar trend to remote
sensing data. As previously noted, MODIS derived estimates of NPP do not fully include the
effect of $CO_2$ fertilization (de Kauwe et al., 2016) whereas ORCHILEAK does. Thus, the
MODIS NPP product may underestimate the increasing trend in NPP, which would bring our
modeled trend further away from this dataset. On the other hand, forest degradation effects and
recent droughts have been associated with a decrease of greenness (Zhou et al., 2014) and
above ground biomass loss (Qie et al., 2019) in tropical forests.
Our results of the historic trend in NEP (not including LUC and harvest fluxes) also generally
concur with other modelling studies of tropical Africa (Fisher et al., 2013). Fisher et al. (2013)
used nine different land surface models to show that the African tropical biome already
represented a natural (i.e. no disturbance, but also neglecting LOAC fluxes) net uptake of
around 50 Tg C $yr^{-1}$ in 1901 and that this uptake more than doubled by 2010. We find a similar
trend though we simulate higher absolute NEP. Indeed, one of the models used in Fisher was
ORCHIDEE and using this model alone, they calculate a virtually identical estimate of net





604 uptake of 277 Tg C yr$^{-1}$ for the present day, though this estimate neglects the transfers of C

605 along the LOAC and would therefore be reduced with their inclusion. Our results also generally

606 concur with estimates based on the upscaling of biomass observations (Lewis et al., 2009).

607 Lewis et al. (2009) up-scaled forest plot measurements to calculate that intact tropical African

608 forests represented a net uptake of approximately 300 Tg C yr$^{-1}$ between 1968 and 2007 and

609 this is consistent with our NEP estimate 275 Tg C yr$^{-1}$ over the same period.

610 Over the entire simulation period (1861-2099), we estimate that aquatic $CO_2$ evasion will

611 increase by 79% and the export of C to the coast by 67%. This increase is considerably higher

612 than the 25% and 30% rise in outgassing and export predicted for the Amazon basin (Lauerwald

613 et al., submitted), over the same period and under the same scenario. This is largely due to the

614 fact climate change is predicted to have a substantial negative impact on the aquatic C fluxes

615 in the Amazon, something that we do not find for the Congo where rainfall is projected to

616 substantially increase over the 21$^{st}$ century (RCP 6.0). In the Amazon, Lauerwald et al.

617 (submitted) show that while there are decadal fluctuations in precipitation and discharge, total

618 values across the basin remain unchanged in 2099 compared to 1861. However, changes in the

619 spatial distribution of precipitation mean that the total water surface area actually decreases in

620 the Amazon. Indeed, while we find an increase in the ratio of C exports to the LOAC/NPP from

621 3 to 5%, Lauerwald et al. (submitted) find a comparative decrease. The increase in the

622 proportion of NPP lost to the aquatic system (Fig. 8, 9) as well as in the concentration of DOC

623 (by 24% at Brazzaville) that we find in the Congo, could have important secondary effects, not

624 least the potential for greater DOC concentrations to cause a reduction in pH levels (Laudon &

625 Buffam, 2008) with implications for the wider ecology (Weiss et al., 2018).

626 Our simulated increase in DOC export to the coast up to the present day is smaller than findings

627 recently published for the Mississippi River using the Dynamic Land Ecosystem Model

628 (DLEM, Ren at al., 2016). In addition, the Mississippi study identified LUC including land



management practices (e.g. irrigation and fertilization), followed by change in atmospheric
$CO_2$, as the biggest factors in the 40% increase in DOC export to the Gulf of Mexico (Ren et
al., 2016). Another recent study (Tian et al., 2015), found an increase in DIC export from
eastern North America to the Atlantic Ocean from 1901-2008 but no significant trend in DOC.
They demonstrated that climate change and increasing atmospheric $CO_2$ had a significant
positive effect on long-term C export while land-use change had a substantial negative impact.
**4.3 Limitations and further model developments**
It is important to note that we can have greater confidence in the historic trend (until present-
day), as the future changes are reliant on the skill of Earth System model predictions and of
course on the accuracy of the RCP 6.0 scenario. There are for example, large uncertainties
associated with the future $CO_2$ fertilization effect (Schimel et al., 2015) and the majority of
land surface models, ORCHILEAK included in its current iteration, do not represent the effect
of nutrient limitation on plant growth meaning that estimates of land C uptake may be too large
(Goll et al., 2017). There are also considerable uncertainties associated with future climate
projections in the Congo basin (Haensler et al., 2013). However, in most cases the future trends
that we find are more or less continuations of the historic trends, which already represent
substantial changes to the magnitude of many fluxes.
Moreover, we do not account for methane fluxes from Congo wetlands, estimated at 1.6 to 3.2
Tg ($CH_4$) per year (Tathy et al., 1992), and instead assume that all C is evaded in the form of
$CO_2$. Another limitation is the lack of accounting for bespoke peatland dynamics in the
ORCHILEAK model. ORCHILEAK is able to represent the general reduction in C
decomposition in water-logged soils and indeed Hastie et al. (2019) demonstrated that
increasing the maximum floodplain extent in the Amazon Basin led to an increase in NEP
despite fueling aquatic $CO_2$ evasion because of the effect of reducing soil heterotrophic





respiration. Furthermore, ORCHILEAK uses a "poor soils" mask forcing file (Fig. 2 j) based
on the Harmonized World Soil Database (FAO/IIASA/ISRIC/ISS-CAS/JRC, 2009), which
prescribes reduced decomposition rates in low nutrient and pH soils (e.g. Podzols and
Arenosols). The effect of the "poor soils" forcing can clearly be seen in the spatial distribution
of the soil C stock in Fig. A3, where the highest C storage coincides with the highest proportion
of poor soils. Interestingly, this does not include the Cuvette Centrale wetlands (Fig. 1), an area
which was recently identified as containing the world's largest intact tropical peatland and a
stock of around 30 Pg C (Dargie at al., 2017). One potential improvement that could be made
to ORCHILEAK would be the development of a new tailored "poor soils" forcing file for the
Congo Basin which explicitly includes Histosols, perhaps informed by the Soil Grids database
(Hengl et al., 2014), to better represent the Cuvette Centrale. This could in turn, be validated
and/or calibrated against the observations of Dargie et al. (2017). A more long-term aim could
be the integration/ coupling of the ORCHIDEE-PEAT module with ORCHILEAK.
ORCHIDEE- PEAT (Qiu et al., 2019) represents peat as an independent sub-grid hydrological
soil unit in which peatland soils are characterized by peat-specific hydrological properties and
multi-layered transport of C and water. Thus far, it has only been applied to northern peatlands,
and calibrating it to tropical peatlands, along with integrating it within ORCHILEAK would
require considerable further model development, but would certainly be a valuable longer-term
aspiration. This could also be applied across the tropical region and would allow us to
comprehensively explore the implications of climate change and land-use change for tropical
peatlands. In addition, ORCHILEAK does not simulate the erosion and subsequent burial of
POC within river and floodplain sediments. Although it does not represent the lateral transfer
of POC, it does incorporate the decomposition of inundated litter as an important source of
DOC and dissolved $CO_2$ to the aquatic system; i.e. it is assumed that POC from submerged
litter decomposes locally in ORCHILEAK. Moreover, previous studies have found that DOC





as opposed to POC (Spencer et al., 2016; Bouillon et al., 2012) overwhelmingly dominates the
total load of C in the Congo. As previously noted, the representation of the rapid C loop of
aquatic macrophytes should also be made a priority in terms of improving models such as
ORCHILEAK, particularly in the tropics. For further discussion of the limitations of
ORCHILEAK, please also see Lauerwald et al. (2017) and Hastie et al. (2019).
**5.   Conclusions**
For the present day, we show that aquatic C fluxes, and in particular $CO_2$ evasion, are important
components of the Congo Basin C balance, larger than for example the combined fluxes from
LUC and harvesting, with around 4% of terrestrial NPP being exported to the aquatic system
each year. We find that these fluxes have undergone considerable perturbation since 1861 to
the present day, and that under RCP 6.0 this perturbation will continue; over the entire
simulation period (1861-2099), we estimate that aquatic $CO_2$ evasion will increase by 79% and
the export of C to the coast by 67%. We further find that the ratio of C exports to the
LOAC/NPP increases from 3 to 5%, driven by both rising atmospheric $CO_2$ concentrations and
climate change.  The increase in the proportion of NPP transferred to the aquatic system (Fig.
8, 9), as well as in the concentration of DOC (by 24% at Brazzaville), could also have important
secondary effects, not least the potential for greater DOC concentrations to cause a reduction
in pH levels (Laudon & Buffam, 2008) with implications for the wider ecology (Weiss et al.,
2018). This calls for long-term monitoring of C levels and fluxes in the rivers of the Congo
basin, and further investigation of the potential impacts of such change, including additional
model developments.

*Code availability*. A description of the general ORCHIDEE code can be found here:
http://forge.ipsl.jussieu.fr/orchidee/browser#tags/ORCHIDEE_1_9_6/ORCHIDEE.





The main part of the ORCHIDEE code was written by Krinner et al. (2005). See d'Orgeval et
al. (2008) for a general description of the river routing scheme. For the updated soil C module
please see Camino Serrano (2015). For the source code of ORCHILEAK see Lauerwald et al.
(2017)- https://doi.org/10.5194/gmd-10-3821-2017-supplement
For details on how to install ORCHIDEE and its various branches, please see the user guide:
http://forge.ipsl.jussieu.fr/orchidee/ wiki/Documentation/UserGuide
*Author contribution*. AH, RL, PR and PC all contributed to the conceptualization of the study.
RL developed the model code, AH developed the novel forcing files for Congo, and AH
performed the simulations. FP provided the GIEMS dataset for model validation. AH prepared
the manuscript with contributions from all co-authors. RL and PR provided supervision and
guidance to AH throughout the research. PR acquired the primary financial support that
supported this research.
*Competing interests*. The authors declare that they have no conflict of interest.
*Financial support*. Financial support was received from the European Union's Horizon 2020
research and innovation programme under the Marie Sklodowska- Curie grant agreement No.
643052 (C-CASCADES project). PR acknowledges funding from the European Union's
Horizon 2020 research and innovation programme under Grant Agreement 776810 (project
VERIFY). RL acknowledges funding from the ANR ISIPEDIA ERA4CS project.

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



*Appendix A*

| Table A 1: Performance statistics for modelled versus observed seasonality of discharge on the Congo at Brazzaville | | | | |
|---|---|---|---|---|
| **Climate forcing** | RSME | NSE | $R^2$ | Mean monthly discharge ($m^3 s^{-1}$) |
| ISIMIP | 29% | 0.20 | 0.23 | 38,944 |
| Princeton GPCC | 40% | -0.25 | 0.20 | 49,784 |
| GSWP3 | 46% | -4.13 | 0.04 | 24,880 |
| CRUNCEP | 65% | -15.94 | 0.01 | 16,394 |
| Observed (HYBAM) | | | | 40,080 |


| Table A 2: Pearson correlation coefficient (r) between detrended carbon fluxes and detrended climate variables | | | | | | | |
|---|---|---|---|---|---|---|---|
| | SHR | Aquatic $CO_2$ evasion | Lateral C | NEP | Rain | Temp. | MEI |
| NPP | -0.48 | 0.68 | 0.72 | 0.90 | 0.64 | -0.57 | -0.09 |
| SHR | | -0.41 | -0.48 | -0.71 | -0.32 | 0.76 | 0.04 |
| Aquatic $CO_2$ evasion | | | 0.92 | 0.41 | 0.87 | -0.30 | -0.21 |
| Lateral C | | | | 0.52 | 0.81 | -0.38 | -0.15 |
| NEP | | | | | 0.40 | -0.74 | -0.01 |
| Rain | | | | | | -0.31 | -0.26 |
| Temp. | | | | | | | 0.03 |



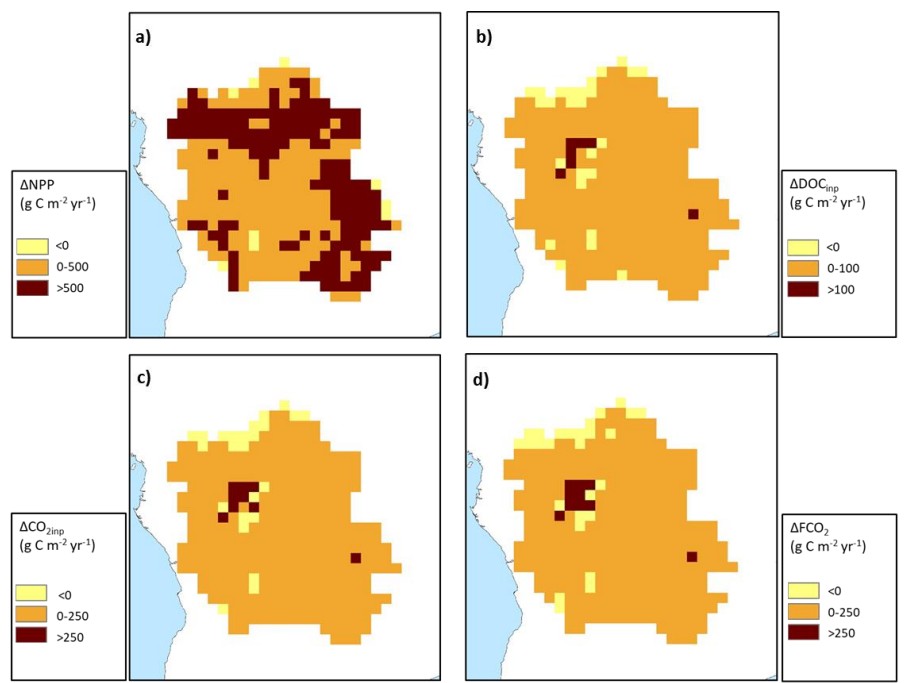


**Figure A 1:Change (Δ, 2099 minus 1861) in the spatial distribution of a) terrestrial NPP, b) DOC leaching into the aquatic system, c) CO₂ leaching into the aquatic system and d) aquatic CO₂ evasion. All at a resolution of 1°**












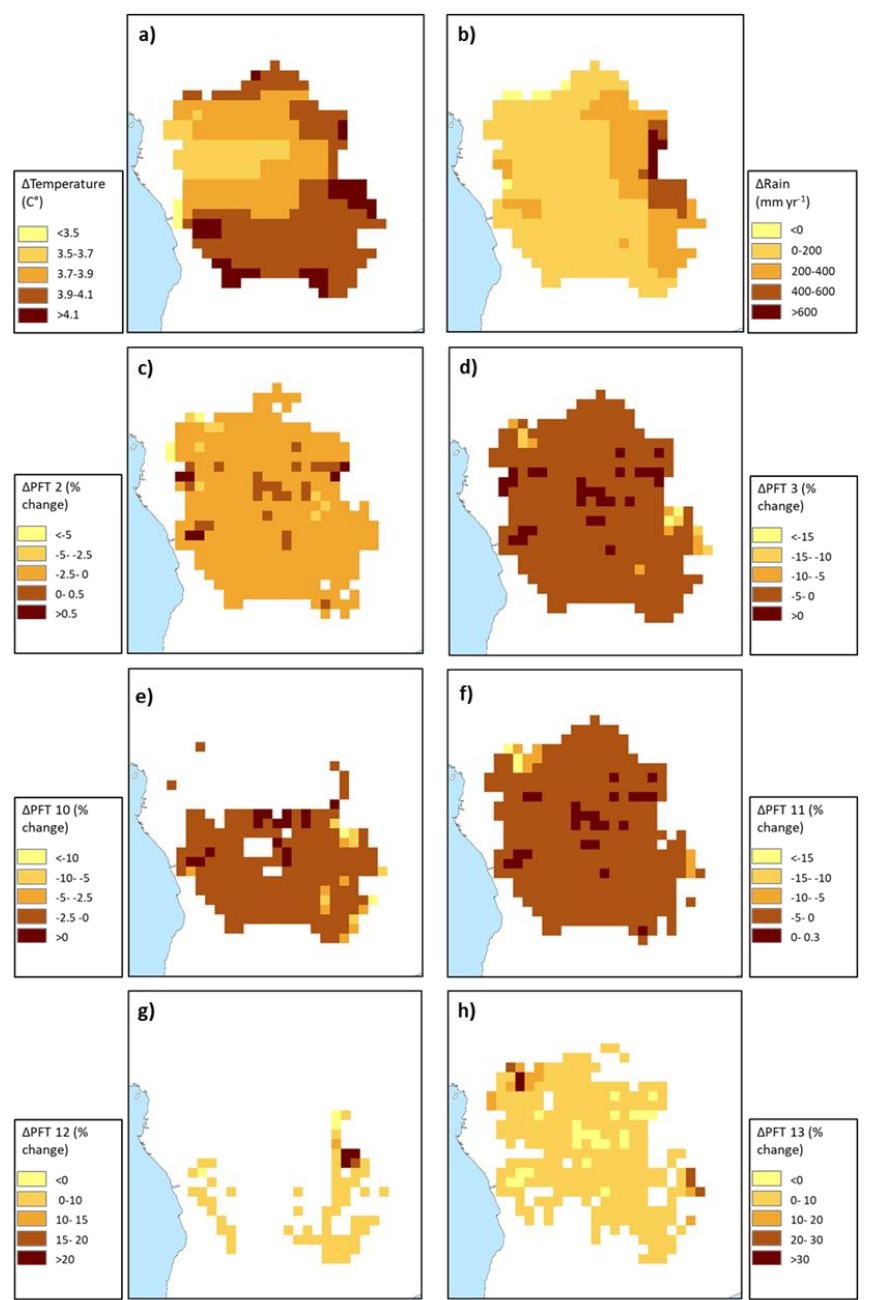

**Figure A 2: Change (Δ, 2099 minus 1861) in the spatial distribution of the principal climate and land-use drivers across the Congo Basin; a) mean annual temperature in °C, b) mean annual rainfall in mm yr⁻¹, c)-h) mean annual maximum vegetated fraction for PFTs 2,3, 10,11,12 and 13. All at a resolution of 1°.**






| Table A 3: Past (1861-1890), present-day (1981-2010) and future (2070-2099) mean values for important climate and land-use drivers across the Congo basin | | | | | | | | |
|---|---|---|---|---|---|---|---|---|
| Period | Temp. | Rain. | PFT2 | PFT3 | PFT10 | PFT11 | PFT12 | PFT13 |
| 1861-1890 | 24.0 | 1451 | 0.263 | 0.375 | 0.154 | 0.254 | 0.015 | 0.014 |
| 1981-2010 | 25.2 | 1526 | 0.255 | 0.359 | 0.154 | 0.255 | 0.038 | 0.030 |
| 2070-2099 | 28.2 | 1654 | 0.258 | 0.362 | 0.147 | 0.245 | 0.039 | 0.037 |

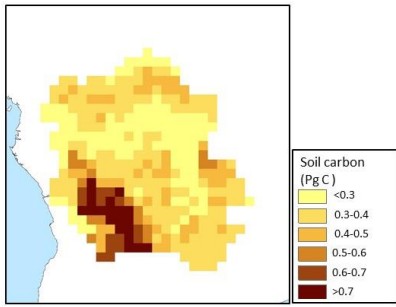

**Figure A 3: Spatial distribution of simulated total carbon stored in soils for the present day (1981-2020).**


