# Peer review of "Historical and future contributions of inland waters to the Congo basin"

_Earth System Dynamics, 2020_

## Referee Comment (RC1) · Anonymous Referee #1 · 25 Mar 2020

GENERAL COMMENTS

Hastie et al. applied the ORCHILEAK model to the Congo Basin to estimate CO2 evasion from the river to the atmosphere, and then provide the long-term (1861-2099) trends of aquatic carbon fluxes. The approach builds on the previous application of the same model to the Amazon by the same group.

The ms fits the scope of ESD, is well written, but the quality of the figures could be improved.

More importantly, a better job should be made at validating the model with a model vs field data comparison of dissolved CO2 concentrations, as detailed below.

Similarly, a better job could be made at discussing the CO2 emissions from the model,

[Figure]

by detailing the model representation of dissolved CO2 concentration, gas transfer velocity and stream surface area, as detailed below.

MAJOR COMMENTS

The model validation step is very slim, as authors compare the model outputs of dissolved organic carbon (DOC) to a sub-set of published field data. DOC in tropical rivers is extremely refractory and provides little grasp on aquatic carbon cycling as the most labile DOC fraction is very rapidly mineralized (both in soils and in water). So DOC provides a poor validation of the carbon cycling in the river, and might be considered almost as a passive tracer and provide a rough validation of the hydrological connectivity between soils and rivers.

Conversely, a convincing validation of the model would be to compare the model outputs of the dissolved CO2 concentration (or the corresponding partial pressure of CO2) with the extensive field data collected by Borges et al. (2019) that are publically available (Borges and Bouillon 2019). While a point by point comparison would not make sense, it could be useful to check if the model captures the overall range of spatial variations along the river and among the different stream sizes. Such validation would be extremely convincing because the CO2 evasion from the river to the atmosphere (the core topic of the paper) is computed from the dissolved CO2 concentration (the atmospheric CO2 is comparatively invariant) and the gas transfer velocity. So if the model does not represent correctly the dissolved CO2 concentration then this implies that the CO2 evasion rates are incorrect (as well as any conclusion based on past reconstruction and future projection from the model outputs).

The overall emission of CO2 from the fluvial component of the Congo Basin (TgC/yr) is based on the product of a CO2 flux density (mol/m2/yr) and a stream surface area; the areal CO2 flux is itself computed from the air-water CO2 concentration gradient, and the gas transfer velocity; the air-water CO2 concentration gradient in turn is mainly function of the dissolved CO2 concentration. So there are three quantities that could

explain the difference between the 4 estimates of integrated CO2emissions discussed in section L-513-517: the CO2 dissolved concentration, the gas transfer velocity and the stream surface area. The evaluation of the model performance would be much more convincing if each of these three quantities was compared to available estimates.

Final elements of validation and discussion that are missing are the contribution of $HCO_3-$ to the export of dissolved inorganic carbon (DIC) from the river to the ocean. The export of DIC from rivers to the ocean is mainly in the form of $HCO_3-$ including the Congo river (Wang et al. 2013). So ignoring $HCO_3-$ would lead to a very substantial under-estimation of DIC export to the ocean. This should be fairly easy to implement with a weathering model and GIS of lithology. This is of course of interest for the topic of paper as weathering intensity is a function of temperature and precipitation that are used by the authors to study the long-term (1861-2099) trends of aquatic carbon fluxes. Further, there are substantial data-sets of total alkalinity (that mainly corresponds to $HCO_3-$) providing spatial (Borges et al. 2019) and seasonal (Wang et al. 2013; Bouillon et al. 2012; 2013) patterns of $HCO_3-$ variability.

Additionally, dissolved CO2 is in thermodynamic equilibrium with $HCO_3-$, so it is required to have some grasp on $HCO_3-$ variability to correctly model dissolved CO2 dynamics, hence, CO2 emissions to the atmosphere.

SPECIFIC COMMENTS

L 24 : It's the increase of air temperature rather than "climate change" in general.

L44 : In this section it's unclear what is meant by "increase" of "net primary productivity" and "storage in tree biomass". A recent study shows that African forests are sinks of carbon on yearly basis, and that the carbon sink is constant in time from the mid-1990's to present (Hubau et al. 2020). So, according to this study, there is no "increase" in NPP as stated but a constant sink. Please clarify.

L74 : The tropical region is also a hotspot of aquatic C cycling due to wetland productivity and wetland carbon inputs to rivers (Abril & Borges 2019), in addition to "terrestrial NPP". In the Amazon river a large fraction of fluvial CO2 emissions to the atmosphere are sustained by wetland inputs (Abril et al. 2014).

L74: The fact that the "tropical region is a hotspot area for inland water C cycling" (as stated) was authoritatively demonstrated by seminal papers such as Richey et al. (2002) and Melack et al. (2004), more than a decade before the recent Lauerwald et al. (2015) work.

L 78-80: there are (at least) two additional elements of context that could be relevant for the introduction:

1) the Raymond et al. (2013) and the Lauerwald et al. (2015) estimates are in fact based on the same initial data-base of pCO2 computed from pH and alkalinity (GLO-RICH) that was extrapolated globally using two different approaches; this illustrates how uncertain these global estimates are, since the resulting values differ by a factor of 3;

2) While the conclusion of Lauerwald et al. (2015) that the majority of CO2 emissions from rivers comes from the tropics is probably correct, field data (used in the global extrapolation) is nearly absent in the tropics. For instance for the Congo River there is only one single data entry in the GLORICH data-set. Most of the data used to develop the statistical model of Lauerwald et al. (2015) come from non-tropical areas such as North America and Scandinavia.

L 244: Borges et al. (2019) report discharge data from the mainstem Congo at Kisangani. So there are additional data-sets to validate the model hydrology.

L513: It could be useful to put into context how these different fluxes were computed. The Raymond et al. (2013) estimate is based on a single pCO2 value (apparently from pH and alkalinity measurements in Pool Malebo) that was extrapolated to the whole basin. The comparison of this single value of pCO2 with the extensive data set reported

by Borges et al. (2019) shows that it is unrealistically low (refer to Supplemental Figure 18). Lauerwald et al. (2015) et al. estimate of pCO2 compares better to the Borges et al. (2019) data-set but still fails to represent the influence of the Cuvette Centrale (refer to Supplemental Figure 18). Please also note that the CO2 estimate for the Congo reported by Borges et al. (2015) was based on exactly the same stream surface area and gas transfer velocity as those used by Raymond et al. (2013), and also showed that the Raymond et al. (2013) estimate was under-estimated (obviously, since the pCO2 value is unrealistically low). So there is some clear convergence that the present estimate of CO2 emission based on ORCHILEAK is under-estimated even if it is coincidentally close to the one reported by Raymond et al. (2013). The actual reasons of the under-estimation need to be explored as suggested in the above Major Comments.

In the discussion of the long-term changes of DOC export, it could useful to mention that there was no measurable difference in DOC flux from the Oubangui (largest tributary of the Congo) between the 1990's (Coynel et al. 2005) and the 2010's as reported and discussed by Bouillon et al. (2012; 2014).

REFERENCES

Abril G et al. (2014) Amazon River carbon dioxide outgassing fuelled by wetlands. Nature 505, 395-398.

Abril G & AV Borges (2019) Carbon leaks from flooded land: do we need to re-plumb the inland water active pipe? Biogeosciences, 16, 769-784, doi: 10.5194/bg-16-769-2019

Borges AV et al. (2015) Globally significant greenhouse-gas emissions from African inland 733 waters. Nature Geoscience, 8, 637. https://doi.org/10.1038/ngeo2486

Borges AV et al. (2019) Variations in dissolved greenhouse gases (CO2, CH4, N2O) in the Congo River network overwhelmingly driven by fluvial-wetland connectivity, Bio-

geosciences, 16, 3801–3834, https://doi.org/10.5194/bg-16-3801-2019.

Borges AV & S Bouillon. (2019) Data-base of CO2, CH4, N2O and ancillary data in the Congo River (Version latest) [Data set]. Biogeosciences. Zenodo. http://doi.org/10.5281/zenodo.3413449

Bouillon S et al. (2012) Organic matter sources, fluxes and greenhouse gas exchange in the Oubangui River (Congo River basin), Biogeosciences, 9, 2045–2062, https://doi.org/10.5194/bg-9-2045-2012

Bouillon S et al. (2014) Contrasting biogeochemical characteristics of the Oubangui River and 748 tributaries (Congo River basin). Scientific Reports, 4, 5402. https://doi.org/10.1038/srep05402

Coynel A et al. (2005) Spatial and seasonal dynamics of total suspended sediment and organic carbon species in the Congo River, Global Biogeochemical Cycles, 19, GB4019, doi:10.1029/2004GB002335

Hubau W et al. (2020) Asynchronous carbon sink saturation in African and Amazonian tropical forests, Nature, 579, 80-87, https://doi.org/10.1038/s41586-020-2035-0

Lauerwald R et al. (2015) Spatial patterns in CO2 evasion from the global river network. Global Biogeochemical Cycles, 29(5), 534-554.

Melack JM et al (2004) Regionalization of methane emissions in the Amazon Basin with microwave remote sensing. Global Change Biology, 10, 530-544

Raymond PA et al. (2013) Global carbon dioxide emissions from inland waters. Nature, 503, 355-359.

Richey JE et al. (2002) Outgassing from Amazonian rivers and wetlands as a large tropical source of atmospheric CO2. Nature, 416, 617-620.

Wang ZA et al. (2013) Inorganic carbon speciation and fluxes in the Congo River, Geophysical Research Letters, 40, doi:10.1002/grl.50160.

**[ESDD](javascript:void(0))**

---

## Referee Comment (RC2) · Anonymous Referee #2 · 18 Apr 2020

Hastie and co-authors application of a sophisticated hydrological/biogeochemical model to the Congo basin is ambitious given the paucity of historical and current data, and large uncertainties in future climates and land uses for the basin. Many aspects of the model and its application raise conceptual and empirical questions. Hence, the veracity of the results would seem much more uncertain than suggested by the manuscript. Before modeling possible changes through the 21st century, it would seem necessary to rigorously evaluate the performance of the model under current conditions. In general, the manuscript would benefit from editorial work on the style, flow and focus.

Abstract The specific numbers for historical and projected fluxes of CO2 and DOC, based on modelled results, should be stated with caveats, not just uncertainty ranges,

given the many, necessary assumptions and systematic issues with modelled results. This is a fundamental problem with extrapolating backwards and forward when the underlying algorithms and forcings all have serious caveats.

Introduction L44-58: Though based on published papers, the estimates of carbon stocks and fluxes in the forests and soils of the Congo would benefit from a more critical evaluation given the logistic difficulties and paucity of data for the region. L68-73: It is important to mention that a considerable portion of carbon being processed in the 'land-ocean aquatic continuum (LOAC)' is derived from NPP within the aquatic systems, not just carbon derived from uplands. L73-74: To support the statement that 'The tropical region is a hotspot area for inland water C cycling' it would be more appropriate to cite results from empirical studies, rather than modelled estimates. L81-82: How well are the current fluxes known? L86-92: These are rather ambitious goals, given the large uncertainties in current conditions and paucity of historical and current data.

Methods ORCHILEAK is a valuable modification to the land surface model, ORCHIDEE, and is well described in Lauerwald et al., 2017. Given that 'All of the processes represented in ORCHILEAK remain identical to those previously represented for the Amazon ORCHILEAK', the veracity of the model for the Amazon would need careful evaluation before accepting its use in the Congo. It is outside the scope of this review to revisit issues, some of which were noted by the authors, with regard the application to the Amazon. However, it is misleading to state that 'ORCHILEAK model . . . is capable of simulating both terrestrial and aquatic C fluxes in a consistent manner for the present day in the Amazon and Lena' without caveats and limitations acknowledged. Moreover, the differences between the Congo and Amazon would seem to require thorough considerable before accepting identical application. As described in Borges et al. (2019): The Congo basin has a wide range of tributaries with differing lithology, soils, vegetation and rainfall in their catchments, has extensive peat deposits, and has large areas of year-round inundation. These conditions differ significantly from

the Amazon basin. L111: Camino Serrano 2015 is not listed in references. In Lauer-wald et al., 2017 this reference is listed as - Camino Serrano, M.: Factors controlling dissolved organic carbon in soils: a database analysis and a model development, Universiteit Antwerpen, Belgium, 2015. This is not readily accessible. L124: Why is the water surface area varied diurnally? Figure 1. The figure needs latitudes and longitudes indicated. Lake Tanganyika is drawn as if a loop of rivers; redraw as a lake. Figure 2 and associated text (L153-168) do not consider the veracity of these data. Though 13 plant functional groups (pft) are prescribed, how well are their ecophysiological characteristics in the conditions of the Congo known? 'Tropical broadleaved raingreen trees' is an odd phrase. Section 2.2: Given the importance of the wetlands to the modeling, further discussion of datasets used is warranted. L177: What is the definition of swamps versus floodplains and how are they distinguished in the Congo? L178: Does inundation of the floodplains require exceedance of 'bank-full discharge'? See comment about section 2.3. L179-180: It is unclear why 'a constant proportion of river discharge is fed into the base of the soil column'. L188-190: Round the MFF to 10%. Is this value the maximum MFF or the mean maximum? L193: How are 'fens' different from swamps in the Congo? Section 2.3: Indeed, simulating the hydrology well is critical. The description of the calibration steps is somewhat confusing. For example, line 217 states 'Without calibration, the majority of the different climate forcing model runs performed poorly . .'. However, key hydrological parameters needed calibration. Hence, it would seem issues with both forcings and model parameters are confounded. L204-206: The comment 'no data is available with which to directly evaluate the simulation of DOC and $CO_2$ leaching from the soil to the river network' is a useful caveat which makes validation of the coupling of uplands and wetlands to the rivers seriously problematic. L232: '95th percentile of water level heights (floodh95th)' would seem to require information about the topography of the area being inundated. L233-240: The concept of bank-full discharge as a threshold for initiation of inundation of floodplains is questionable as applied to tropical floodplain such as those in the Amazon or Congo. Studies inundation dynamics in the Amazon with detailed measurements or modeling

indicate that inundation occurs more or less continuously as the rivers rise and that the water comes from both the rivers and uplands (e.g., Lesack and Melack 1995 Water Resources Res 31:329–334; Bonnet et al. 2017 Hydrol. Processes 31: 1702–1718; Rudorff et al 2014 Water Resources Res 31:329–349; Ji et al. 2019 Water Resources Res 54). L248-249: The algorithms used to generate the GIEMS vary in their effectiveness depending the density and extent of the inundated vegetation. Section 2.4.1: How well do the soil processes derived for Europe (Camino Serrano et al. 2018) apply to the Congo, how were the passive, slow and active pools determined and how were the decomposition rates in the flooded and non-flooded soil derived? Section 2.4.2: What were the projected land use changes? These would seem rather difficult to prescribe, as noted in the text. The exclusion of shifting cultivation would seem a serious omission. Section 2.6: The terms in Equation 1 would all seem to be quite difficult to calculate and to validate.

Results Section 3.1: In general, simulations of mean monthly discharge for large tropical river systems without large dams at downstream stations has been demonstrated as feasible with several models. Hydrological simulations can become increasingly difficult as the scale decreases, as indicated by the less successful simulations of the Ubangi River. Though the text comparing the GIEMS and simulated inundated areas makes sense, the issue of topography as a factor influencing simulated inundated area deserves mention. L358-362: These judgments should be left to the reader to make.

Section 3.2: What is the basis for the calculated standard deviations for the fluxes? Figure 5 would be clearer if redrafted larger with simpler graphics. Given all the uncertainties in the modeling and underlying data, Figure 6 would seem quite questionable.

Section 3.3: These results seem premature without a thorough, rigorous evaluation of the model's output under current conditions.

Section 3.4: 'The dramatic increase in the concentration of atmospheric $CO_2$ (Fig. 8 g) and subsequent fertilization effect on terrestrial NPP has the greatest overall impact on

all of the fluxes across the simulation period' is a critical point and raises a fundamental question about the veracity of the projected changes. As illustrated in a recent paper (Jiang et al. 2020 Nature 580:227-231), the possible CO2 enrichment effects on mature forests are not well captured by current models and need considerably more work to be understood and properly incorporated into models. Figure 9 would be clearer if redrafted larger with simpler graphics. The colors and simple depictions of habitats are distractions.

Discussion Section 4.1: It is not clear that CO2 enrichment effects on photosynthesis results in enhancement of NPP. Though the comparisons of modeled results with regional estimates of biomass and soil C stocks seem reasonable, the empirical estimates have considerable methodological and sampling uncertainty. L500-502: That the CO2 evasion from the water surfaces is sustained by leaching of dissolved CO2 and DOC from soils is not established. In situ C fixation by wetlands and subsequent decomposition of this material could be a significant source of the CO2 evaded as suggested by Borges, and Abril for the Amazon. Indeed, in lines 530-555, the authors discuss the likely contribution of aquatic macrophytes to the available C, and duly note the difficulty of incorporating these plants into their model. However, it is therefore odd that this possible contribution is then discounted in lines 555 to 560. L537-539: It is not correct that strong currents limit the abundance of aquatic macrophytes in the Amazon since most of their growth occurs on floodplains where they can cover large areas. L570-572: Both these estimates of the % of NPP per year transferred to inland waters are based on the same model. What are the estimates for the Amazon based on empirical data? L572-582: This discussion of differences between the Amazon and Congo is too simplistic and not representative of the relevant conditions in either system. It would best be deleted unless considerable more information is added.

Section 4.2: As noted above, it seems a real stretch to be projecting through the 21st century. L610-625: As this section is written as a comparison with Lauerwald et al. (submitted), it does seem suitable to include until Lauerwald et al is available. Also,
there are publications that project hydrological and land use changes in the Amazon. L626-624: This paragraph does not seem necessary since these systems are quite different from the Congo and other examples could be selected.

Section 4.3: Lines 636-645 re-enforce the issues raised above regarding the projections through the 21st century and the question of whether their inclusion in this paper is warranted.

Conclusion L692-696: Is it likely that an increase in DOC from 9.5 to 11.5 mg C/L will cause ecologically meaningful changes in pH?

---

## Author Comment (AC1) · 25 May 2020

Response to reviewer 1 (response in red).

We thank Reviewer 1 for taking the time to review the paper. We agree that additional validation, of pCO2 would improve the paper, as well as some additional analysis of the causes for the discrepancy between our estimates of riverine FCO2 vs those of Borges et al (2019), such as gas transfer velocity and stream surface area. We respond in more detail below.

The model validation step is very slim, as authors compare the model outputs of dissolved organic carbon (DOC) to a sub-set of published field data. DOC in tropical rivers is extremely refractory and provides little grasp on aquatic carbon cycling as the most labile DOC fraction is very rapidly mineralized (both in soils and in water). So DOC provides a poor validation of the carbon cycling in the river, and might be considered almost as a passive tracer and provide a rough validation of the hydrological connectivity between soils and rivers. Conversely, a convincing validation of the model would be to compare the model outputs of the dissolved CO2 concentration (or the corresponding partial pressure of CO2) with the extensive field data collected by Borges et al. (2019) that are publically available (Borges and Bouillon 2019). While a point by point comparison would not make sense, it could be useful to check if the model captures the overall range of spatial variations along the river and among the different stream sizes. Such validation would be extremely convincing because the CO2 evasion from the river to the atmosphere (the core topic of the paper) is computed from the dissolved CO2 concentration (the atmospheric CO2 is comparatively invariant) and the gas transfer velocity. So if the model does not represent correctly the dissolved CO2 concentration then this implies that the CO2 evasion rates are incorrect (as well as any conclusion based on past reconstruction and future projection from the model outputs).

Thank you for these suggestions. We have conducted some analysis (in line with your suggestions) of the monthly average pCO2 over the decade from 2005-2014, and this shows that we are able to capture well, the differences in pCO2 along the mainstem (and their seasonal variations) outlined in Borges et al (2009). Note that we analysed the decadal average to try to minimise the influence of interannual variation for a given year.

For example, during highwater (mean of 6 consecutive months of highest flow, 2005-2014) we simulate a mean pCO2 of 3,373 ppm and 5,007 ppm at Kinsangani and Kinshasha respectively, compared to Borges et al's values of 2,424 ppm and 5,343 ppm during highwater (measured in December 2013).

Similarly, during low flow season (mean of 6 consecutive months of lowest flow, 2005-2014) we simulate a mean pCO2 of 1,568 ppm and 2,643 ppm at Kinsangani and Kinshasha respectively, compared to Borges et al's values of 1,670 ppm and 2,896 ppm during falling water (June 2014).

Moreover, for the Oubangui river at Bangui we simulate a range (2005-2014) of 402 ppm to 3,195 ppm across months, similar to the seasonal variation of 470 to 3,750 observed by Bouillon et al. (2012) over 2010-2011.

However, our simulations fail to capture the very high mean pCO2 values (i.e. up to 12,000 ppm) observed in the smallest tributaries of the Congo in Borges et al. (2019). The highest mean riverine (in headwaters- represented by the fast reservoir) pCO2 values that we simulate for the present day are around 9,000 ppm.  In ORCHILEAK, note that for the fast reservoir we assume a full pCO2 equilibrium with the atmosphere over one full day, which prevents very high pCO2 values from building.

As discussed in the paper, the lack of accounting for Aquatic Macrophytes is likely to be one reason for our relatively low CO2 evasion from the river surface, amongst others covered in the discussion. However, as noted in the manuscript we estimate a similar overall CO2 evasion to Borges et al (2019) once we include the flux from floodplains/ wetlands, which account for 85% of our total CO2 evasion. Moreover, the majority of this flux comes from the Cuvette Centrale (Fig 6) which suggests that while ORHILEAK fails to attribute the majority of this to small rivers (owing to the coarse resolution of the river network in the model at 0.5 degree ~ 50 km) we nonetheless do capture the source of carbon. In other words, in ORCHILEAK the majority of this carbon evades directly from the floodplain and wetlands of the Cuvette Centrale as opposed to the rivers.

In the revised manuscript we would elaborate further on this pCO2 validation in the results section, and add 1 or possibly 2 new figures based on these results. We would also modify the discussion to reflect these new insights.

The overall emission of CO2 from the fluvial component of the Congo Basin (TgC/yr) is based on the product of a CO2 flux density (mol/m2/yr) and a stream surface area; the areal CO2 flux is itself computed from the air-water CO2 concentration gradient, and the gas transfer velocity; the air-water CO2 concentration gradient in turn is mainly function of the dissolved CO2 concentration. So there are three quantities that could explain the difference between the 4 estimates of integrated CO2emissions discussed in section L-513-517: the CO2 dissolved concentration, the gas transfer velocity and the stream surface area. The evaluation of the model performance would be much more convincing if each of these three quantities was compared to available estimates.

The only study of the Congo which we know of where k600 has been measured (albeit, indirectly) is Borges et al. (2015, Nature) based on measurements FCO2 and pCO2. They calculate an average k600 for the Congo river of 2.9 m d-1, relatively similar to our riverine value used of 3.5 m d-1, which we discuss in line 519. Note also that the sensitivity analysis performed in Lauerwald et al. (2017) showed that in the physical approach of ORCHILEAK, CO2 evasion is not very sensitive to the k value, unlike data-driven models. Namely, Lauerwald et al (2017) showed that an increase or decrease of k600 for rivers and swamps of 50% only led to 1% and -4% change in total CO2 evasion, respectively. Rather, ORCHILEAK is sensitive to CO2 inputs from soils and floodplains, as well as instream production through DOC decomposition.

In terms of surface area, we simulate a mean present day River surface area of 25,900 km2 compare to 23,190  km2 (3% of 773 000 km$^2$ – we assume the 3% is a rounded figure so this is likely larger) in Borges et al. (2019, see section 3.8) or the 26,517 km2 quoted in Borges et al (2015, Scientific Reports). Therefore, we think that this can be discounted as a major reason for the discrepancy and we will reiterate this in the revised manuscript.

Final elements of validation and discussion that are missing are the contribution of HCO3- to the export of dissolved inorganic carbon (DIC) from the river to the ocean. The export of DIC from rivers to the ocean is mainly in the form of HCO3- including the Congo river (Wang et al. 2013). So ignoring HCO3- would lead to a very substantial under-estimation of DIC export to the ocean. This should be fairly easy to implement with a weathering model and GIS of lithology. This is of course of interest for the topic of paper as weathering intensity is a function of temperature and precipitation that are used by the authors to study the long-term (1861-2099) trends of aquatic carbon fluxes. Further, there are substantial data-sets of total alkalinity (that mainly corresponds to HCO3-) providing

spatial (Borges et al. 2019) and seasonal (Wang et al. 2013; Bouillon et al. 2012; 2013) patterns of $HCO_3^-$ variability. Additionally, dissolved $CO_2$ is in thermodynamic equilibrium with $HCO_3^-$, so it is required to have some grasp on $HCO_3^-$ variability to correctly model dissolved $CO_2$ dynamics, hence, $CO_2$ emissions to the atmosphere.

We appreciate these comments and suggestions and acknowledge that the lack of accounting for $HCO_3^-$ is a limitation of the model. However, we would consider creating and implementing a weathering model outside the scope of our study and therefore propose that as alternative, we add estimates from the literature and discuss these limitations with additional caveats.

We estimate a free dissolved $CO_2$ export to the coast of 2.4 Tg C yr-1 for the present day, relatively similar to the empirically derived estimate of the total DIC export of 3.3 Tg C yr-1 calculated in Wang et al (2013). According to Wang et al, dissolved $CO_2$ accounts for the majority (1.9 Tg C yr-1) but the difference between our estimate and theirs is indeed likely to be largely caused by our lack of accounting for the weathering derived flux ($HCO_3^-$) which they estimate at 1.4 Tg C yr-1. In summary, despite these limitations the results of Wang et al, suggest that we still capture the majority of the DIC flux.

However, Cai et al. (2008, Continental Shelf Research), suggest that the total DIC flux could potentially be substantially larger at around 13 Tg C yr-1. We will discuss these issues in more detail in the revised manuscript.

SPECIFIC COMMENTS L 24 : It's the increase of air temperature rather than "climate change" in general.

Thanks, will change as suggested

L44 : In this section it's unclear what is meant by "increase" of "net primary productivity" and "storage in tree biomass". A recent study shows that African forests are sinks of carbon on yearly basis, and that the carbon sink is constant in time from the mid-1990's to present (Hubau et al. 2020). So, according to this study, there is no "increase" in NPP as stated but a constant sink. Please clarify.

We will modify these sentences in line with the new findings of Hubau et al. 2020. This paper was published after we submitted and previous literature indicated that there was an increase in tree biomass (Lewis et al., 2009) between 1968- 2007. By tree biomass we mean the biomass and in turn carbon storage (Mg C ha$^{-1}$ yr$^{-1}$) of intact forests (as surveyed by the team of Lewis et al). Please also see response to Reviewer 2.

L74 : The tropical region is also a hotspot of aquatic C cycling due to wetland productivity and wetland carbon inputs to rivers (Abril & Borges 2019), in addition to "terrestrial NPP". In the Amazon river a large fraction of fluvial $CO_2$ emissions to the atmosphere are sustained by wetland inputs (Abril et al. 2014). L74: The fact that the "tropical region is a hotspot area for inland water C cycling" (as stated) was authoritatively demonstrated by seminal papers such as Richey et al. (2002) and Melack et al. (2004), more than a decade before the recent Lauerwald et al. (2015) work. L 78-80: there are (at least) two additional elements of context that could be relevant for the introduction: 1) the Raymond et al. (2013) and the Lauerwald et al. (2015) estimates are in fact based on the same initial data-base of pCO2 computed from pH and alkalinity (GLORICH) that was extrapolated globally using two different approaches; this illustrates how uncertain these global estimates are, since the resulting values differ by a factor of 3; 2) While the conclusion of Lauerwald et al. (2015) that the majority of $CO_2$ emissions from rivers comes from the tropics is probably correct, field data (used in

the global extrapolation) is nearly absent in the tropics. For instance for the Congo River there is only one single data entry in the GLORICH data-set. Most of the data used to develop the statistical model of Lauerwald et al. (2015) come from non-tropical areas such as North America and Scandinavia.

We will modify the introduction in line with these suggestions, prioritising empirically derived estimates wherever possible, and add additional context.

L 244: Borges et al. (2019) report discharge data from the mainstem Congo at Kisangani. So there are additional data-sets to validate the model hydrology.

We performed the Hydrology validation before this paper was published but would be happy to validate our simulation against this additional data source, or alternatively could change the wording of this sentence.

L513: It could be useful to put into context how these different fluxes were computed. The Raymond et al. (2013) estimate is based on a single $pCO_2$ value (apparently from pH and alkalinity measurements in Pool Malebo) that was extrapolated to the whole basin. The comparison of this single value of $pCO_2$ with the extensive data set reported by Borges et al. (2019) shows that it is unrealistically low (refer to Supplemental Figure 18). Lauerwald et al. (2015) et al. estimate of $pCO_2$ compares better to the Borges et al. (2019) data-set but still fails to represent the influence of the Cuvette Centrale (refer to Supplemental Figure 18). Please also note that the $CO_2$ estimate for the Congo reported by Borges et al. (2015) was based on exactly the same stream surface area and gas transfer velocity as those used by Raymond et al. (2013), and also showed that the Raymond et al. (2013) estimate was under-estimated (obviously, since the $pCO_2$ value is unrealistically low). So there is some clear convergence that the present estimate of $CO_2$ emission based on ORCHILEAK is under-estimated even if it is coincidentally close to the one reported by Raymond et al. (2013). The actual reasons of the under-estimation need to be explored as suggested in the above Major Comments.

We agree that some additional context would be helpful to the reader, as well as the additional validation (of $pCO_2$ etc) which we cover in our response to your major comments.

In the discussion of the long-term changes of DOC export, it could useful to mention that there was no measurable difference in DOC flux from the Oubangui (largest tributary of the Congo) between the 1990's (Coynel et al. 2005) and the 2010's as reported and discussed by Bouillon et al. (2012; 2014).

Thank you for pointing this out and we will add a sentence on this.

References

Borges, A. V, Darchambeau, F., Teodoru, C. R., Marwick, T. R., Tamooh, F., Geeraert, N., Bouillon, S. (2015). Globally significant greenhouse-gas emissions from African inland waters. Nature Geoscience, 8, 637. Retrieved from https://doi.org/10.1038/ngeo2486

Borges, A. V., Darchambeau, F., Lambert, T., Morana, C., Allen, G. H., Tambwe, E., Toengaho Sembaito, A., Mambo, T., Nlandu Wabakhangazi, J., Descy, J.-P., Teodoru, C. R., and Bouillon, S (2019).: Variations in dissolved greenhouse gases (CO2, CH4, N2O) in the Congo River network

overwhelmingly driven by fluvial-wetland connectivity, Biogeosciences, 16, 3801–3834, https://doi.org/10.5194/bg-16-3801-2019.

Cai, W.-J., Guo, X., Chen, C. T. A., Dai, M., Zhang, L., Zhai, W., Lohrenz, S. E., Yin, K., Harrison, P. J., and Wang, Y.: A comparative overview of weathering intensity and HCO− 3 flux in the world's major rivers with emphasis on the Changjiang, Huanghe, Zhujiang (Pearl) and Mississippi Rivers, Cont. Shelf Res., 28, 1538–1549, 2008.

Lauerwald, R., Regnier, P., Camino-Serrano, M., Guenet, B., Guimberteau, M., Ducharne, A., … Ciais, P. (2017). ORCHILEAK (revision 3875): a new model branch to simulate carbon transfers along the terrestrial--aquatic continuum of the Amazon basin. *Geoscientific Model Development*, *10*(10), 3821–3859. https://doi.org/10.5194/gmd-10-3821-2017

Lewis, S. L., Lopez-Gonzalez, G., Sonké, B., Affum-Baffoe, K., Baker, T. R., Ojo, L. O., … Wöll, H. (2009). Increasing carbon storage in intact African tropical forests. Nature, 457, 1003. Retrieved from https://doi.org/10.1038/nature07771

Wang, Z. A., Bienvenu, D. J., Mann, P. J., Hoering, K. A., Poulsen, J. R., Spencer, R. G. M., and Holmes, R. M. ( 2013), Inorganic carbon speciation and fluxes in the Congo River, *Geophys. Res. Lett.*, 40, 511– 516, doi:10.1002/grl.50160.

---

## Author Comment (AC2) · 25 May 2020

Response to reviewer 2 (response in red).

We thank Reviewer 2 for taking the time to review the paper. We believe that the majority of the comments from reviewer 2 can be addressed relatively easily through changes to the text, namely additional caveats and discussion, combined with more tentative language (especially for sections 3.3. and 3.4).

As noted by the reviewer, the most uncertain aspect of our results is the future projections. However, it is largely very recent research (Hubau et al.,2020; Jiang et a., 2020 both in Nature and published after the submission of our paper) which brings our future projections, specifically the effect of $CO_2$ fertilization into some doubt. Indeed, these findings could lead to no less than a paradigm shift in the current generation of land-surface models. Note that until recently, field observations (Lewis et al., 2009, from 1968 and 2007) largely supported our results

Rather than removing this section altogether, we offer to keep our full results while adding further discussion and caveats, centred around these new papers. We feel that while future projections of NPP in response to rising $CO_2$ are highly uncertain in tropical forests where there is no elevated $CO_2$ experiment (none published thus far, though an Amazon FACE experiment is underway) and where nutrient limitations may decrease $CO_2$ induced enhancements, the increase in the export of carbon to the LOAC and coast (the main/ highlighted result of the future projections) remains an interesting result, and one which is also due to other drivers such as climate change. As such we still feel that these results merit inclusion and discussion, albeit with additional caveats/ discussion reiterating the uncertainty of future projections, and centred around these new publications. We also propose that we change some of the language in this section of the paper (as well as Abstract) to be more tentative, to reflect these new publications. Please see response to detailed comments below.

Introduction L44-58: Though based on published papers, the estimates of carbon stocks and fluxes in the forests and soils of the Congo would benefit from a more critical evaluation given the logistic difficulties and paucity of data for the region.

We take this point and will provide a more critical evaluation of the current published estimates

L68- 73: It is important to mention that a considerable portion of carbon being processed in the 'land-ocean aquatic continuum (LOAC)' is derived from NPP within the aquatic systems, not just carbon derived from uplands.

We will modify the manuscript in line with this comment

L73-74: To support the statement that 'The tropical region is a hotspot area for inland water C cycling' it would be more appropriate to cite results from empirical studies, rather than modelled estimates.

We will modify these lines to cite results from empirical studies over modelled where possible such as Richey et al. (2002, Nature), Rasera et al (2013, Biogeochemistry), Abril et al (2014, Nature) and the various papers by Borges et al.

L81-82: How well are the current fluxes known?

There are still considerable uncertainties associated with the current fluxes and will change wording to reflect this, adding uncertainty ranges where possible.

L86-92: These are rather ambitious goals, given the large uncertainties in current conditions and paucity of historical and current data.

We agree that these are ambitious goals given the uncertainties and paucity of data. However, we would argue that it is still better to present the full results, but with the caveats up front (including in abstract) along with uncertainty ranges.

Methods ORCHILEAK is a valuable modification to the land surface model, ORCHIDEE, and is well described in Lauerwald et al., 2017. Given that 'All of the processes represented in ORCHILEAK remain identical to those previously represented for the Amazon ORCHILEAK', the veracity of the model for the Amazon would need careful evaluation before accepting its use in the Congo. It is outside the scope of this review to revisit issues, some of which were noted by the authors, with regard the application to the Amazon. However, it is misleading to state that 'ORCHILEAK model . . . is capable of simulating both terrestrial and aquatic C fluxes in a consistent manner for the present day in the Amazon and Lena' without caveats and limitations acknowledged.

We accept the point that it is a misleading sentence without caveats, and will acknowledge the caveats and limitations within these sentences.

Moreover, the differences between the Congo and Amazon would seem to require thorough considerable before accepting identical application. As described in Borges et al. (2019): The Congo basin has a wide range of tributaries with differing lithology, soils, vegetation and rainfall in their catchments, has extensive peat deposits, and has large areas of year-round inundation. These conditions differ significantly from the Amazon basin.

We accept that the conditions in the Amazon and Congo are very different, though the Amazon also has been shown to contain significant peat deposits-( see for example Draper et al, 2014 and is expected to have larger 'undiscovered peatlands' Gumbricht et al., 2017), and also a wide range of tributaries with differing lithology soils etc, as well as a large east-west precipitation gradient. We would debate the term 'identical application' as we recalibrated the model as fully as we could with the available data, under the current model structure (admittedly with associated limitations and caveats).

L111: Camino Serrano 2015 is not listed in references. In Lauerwald et al., 2017 this reference is listed as - Camino Serrano, M.: Factors controlling dissolved organic carbon in soils: a database analysis and a model development, Universiteit Antwerpen, Belgium, 2015. This is not readily accessible.

Thanks for pointing out. The corrected reference is Camino Serrano et al (2018, GMD). Noted and will change accordingly

L124: Why is the water surface area varied diurnally?

This is the time step for the routing scheme of ORCHILEAK and the surface area varies with discharge.

Figure 1. The figure needs latitudes and longitudes indicated. Lake Tanganyika is drawn as if a loop of rivers; redraw as a lake.

We will add lat and long, and modify the representation of Lake Tanganyika

Figure 2 and associated text (L153-168) do not consider the veracity of these data. Though 13 plant functional groups (pft) are prescribed, how well are their ecophysiological characteristics in the

conditions of the Congo known? 'Tropical broadleaved raingreen trees' is an odd phrase. Section 2.2: Given the importance of the wetlands to the modeling, further discussion of datasets used is warranted.

We take these points and will add some evaluation of the data veracity where possible, for example in comparison with Haensler et al., 2013.

L177: What is the definition of swamps versus floodplains and how are they distinguished in the Congo?

In terms of determining the maximum extent of swamps and floodplains forcing files, these are taken from the dataset of Gumbricht et al. (2017) and therefore follow his definitions. His definition of swamps can be found in Table 1 of his paper but the main characteristics are as follows: "Usually bound to valleys and plains; planar surfaces. Wet all year around, but not necessarily inundated. Usually tree covered." This will be clarified in the revised manuscript.

The max floodplain is defined by aggregating all of the wetland categories in the Gumbricht dataset (including swamps).

In terms of the representation in ORCHILEAK, the difference between swamps and floodplains is outlined in section 2.2. See Lauerwald et al. (2017) for further details.

L178: Does inundation of the floodplains require exceedance of 'bank-full discharge'? See comment about section 2.3.

Yes it does, and bank-full discharge is defined as the median stream flow over the period 1990 to 2005.

L179-180: It is unclear why 'a constant proportion of river discharge is fed into the base of the soil column'.

ORCHILEAK does not yet explicitly represent a groundwater reservoir. This imitates how rivers and swamps are hydrologically coupled through the groundwater table. This will be clarified in revised manuscript.

Please see section 2.1.2 and Figure 3 of Lauerwald et al. (2017) for a more detailed explanation.

L188-190: Round the MFF to 10%. Is this value the maximum MFF or the mean maximum?

Mean maximum

L193: How are 'fens' different from swamps in the Congo?

We have merged the swamps and fens categories from Gumbricht et al. (2017) so effectively they are not different in our study. Irrespectively, according to the Gumbicht dataset there are virtually no fens in the Congo. This will be clarified in the revised manuscript.

Section 2.3: Indeed, simulating the hydrology well is critical. The description of the calibration steps is somewhat confusing. For example, line 217 states 'Without calibration, the majority of the different climate forcing model runs performed poorly .'. However, key hydrological parameters needed calibration. Hence, it would seem issues with both forcings and model parameters are confounded.

Virtually all hydrological models require calibration through the modification of model parameters. Admittedly, the forcing datasets generally do not perform as well for the Congo as for the Amazon for example (likely a result of more climate data being available for the gridded climate forcing fields in the Amazon), which is why we tested several different climate forcing data to estimate uncertainties. The performance without calibration would not have been acceptable/ reasonable, and we feel that we stuck the right balance between improving river flow simulation and over-calibration of parameters, keeping in mind the limitations of climate forcing datasets, and that we are calibrating/ validating for 3 quite different situations (the main stem of the Congo, a much smaller tributary, and overall inundation area).

We will make the explanation of the calibration clearer in the manuscript.

L233-240: The concept of bank-full discharge as a threshold for initiation of inundation of floodplains is questionable as applied to tropical floodplain such as those in the Amazon or Congo. Studies inundation dynamics in the Amazon with detailed measurements or modeling indicate that inundation occurs more or less continuously as the rivers rise and that the water comes from both the rivers and uplands (e.g., Lesack and Melack 1995 Water Resources Res 31:329–334; Bonnet et al. 2017 Hydrol. Processes 31: 1702–1718; Rudorff et al 2014 Water Resources Res 31:329–349; Ji et al. 2019 Water Resources Res 54).

While we appreciate this point, we would respectfully debate some of your conclusions. For example, in their paper Bonnet et al. (2017) conclude that "The mainstream was the main input of water to the flooded area, accounting on average for 93% of total water inputs by the end of the water year. Direct precipitation and runoff from uplands contributed less than or equal to 5% and 10%, respectively. The seepage contribution was less than 1%".  They go on to explain that in their model "Diffusive overbank flows occur where the mainstream water level is above levee crests."

Similarly, Rudorff et al. (2014) conclude that "Diffuse overbank flows represent 93% of total river to floodplain discharge"

It is true that Lesack and Melack (1995) find a much higher percentage of inflow coming from runoff (57%) but this is the results from a single case study of a small lake in the central Amazon basin.

Moreover, the majority of the wetlands which we represent in the Congo in ORCHILEAK are swamps, and so do not rely on overtopping at bank-full discharge.

 L248-249: The algorithms used to generate the GIEMS vary in their effectiveness depending the density and extent of the inundated vegetation. Section 2.4.1: How well do the soil processes derived for Europe (Camino Serrano et al. 2018) apply to the Congo, how were the passive, slow and active pools determined and how were the decomposition rates in the flooded and non-flooded soil derived?

One of the main limitations which Camino Serrano et al. (2018) identified with the potential application of ORCHIDEE-SOM to the tropics was the lack of representation of DOC coming from throughfall, which is incorporated into ORCHILEAK (Lauerwald et al., 2017). The other main limitation in applying it to the Congo is the lack of an explicit representation of peatlands which we discuss in detail (see lines 646 to 682).

The active passive and slow pools are explained in detail on page 3832 of Lauerwald et al. (2017) but the main part of the text is as follows " The soil carbon module distinguishes 3 different pools of DOC depending on the source material: active, slow and passive (Camino Serrano et al, 2018 - GMD). The DOC derived from the active SOC pool and metabolic litter is assigned to the active DOC pool, while the DOC derived from the slow and passive SOC pools are assigned to the slow and passive DOC pools, respectively (Eqs. 43–45). A part of DOC derived from structural plant litter, which is related to the lignin structure of the litter pool (Krinner et al., 2005), is allocated to the slow DOC pool, while the remainder feeds the active DOC pool. The proportion of the decomposed litter and SOC that is transformed into DOC instead of $CO_2$ depends on the carbon use efficiency (CUE), set here to a value of 0.5 (Manzoni et al., 2012). Taken that the same residence time for the slow and passive DOC pools is used in ORCHIDEE-SOM (Camino Serrano, 2015), we merge these two pools when computing throughfall and lateral transport of DOC. Thus, the labile pool is identical to the active pool of the soil carbon module, while the refractory pool combines the slow and passive pools. The labile ($FTF,DOClab$ ) and refractory ($FTF,DOCref$ ) proportions of throughfall DOC are added to the active and slow DOC pools of the first soil layer, respectively" We acknowledge the fact that these modelled SOC pools are not measurable, as in any land surface model, and there is no sufficient radiocarbon age data in Congo to accurately calibrate SOC turnovers in the model.

Moreover, note that in ORCHILEAK decomposition rates of SOC, DOC and litter in flooded soils are 3x lower that of those in non- flooded soils. This is based on the findings by Rueda-Delgado et al., (2006) but also supported by additional research such as Dos Santos & Nelson., (2013).

Section 2.4.2: What were the projected land use changes? These would seem rather difficult to prescribe, as noted in the text. The exclusion of shifting cultivation would seem a serious omission.

The main land-use changes are detailed in Figure A2 of the Appendix. We acknowledge the fact that exclusion of shifting cultivation is a major limitation, though one which would be difficult to incorporate in view of the lack of a spatially explicit dataset. The LUH1 reconstruction indicates for instance shifting cultivation affecting all the tropics with a residence time of agriculture of 15 years, whereas the review from Heininan et al. 2017 (Plos one) revised downwards the area of this type of agriculture, with generally low values in Congo, except in the North east and South East, but suggested a shorter turnover of agriculture of two years only. In view of such uncertainties, we did not include shifting agriculture in the model. But added in the discussion the possibility to improve this situation using new remote sensing datasets on high resolution land cover change (Tyukavina et al. 2018, Sci. Adv)

Results Section 3.1: In general, simulations of mean monthly discharge for large tropical river systems without large dams at downstream stations has been demonstrated as feasible with several models. Hydrological simulations can become increasingly difficult as the scale decreases, as indicated by the less successful simulations of the Ubangi River. Though the text comparing the GIEMS and simulated inundated areas makes sense, the issue of topography as a factor influencing simulated inundated area deserves mention. L358-362: These judgments should be left to the reader to make.

We accept these points and will revise and remove these sentences accordingly.

Section 3.2: What is the basis for the calculated standard deviations for the fluxes? Figure 5 would be clearer if redrafted larger with simpler graphics. Given all the uncertainties in the modeling and underlying data, Figure 6 would seem quite questionable.

The standard deviation represents the interannual variation across the relevant period (for example 1981-2010. We agree that the results depicted in Figure 6 have large associated uncertainties and therefore will either remove it or make caveats/ uncertainties clearer.

Section 3.3: These results seem premature without a thorough, rigorous evaluation of the model's output under current conditions. Section 3.4: 'The dramatic increase in the concentration of atmospheric CO2 (Fig. 8 g) and subsequent fertilization effect on terrestrial NPP has the greatest overall impact on all of the fluxes across the simulation period' is a critical point and raises a fundamental question about the veracity of the projected changes. As illustrated in a recent paper (Jiang et al. 2020 Nature 580:227-231), the possible CO2 enrichment effects on mature forests are not well captured by current models and need considerably more work to be understood and properly incorporated into models. Figure 9 would be clearer if redrafted larger with simpler graphics. The colors and simple depictions of habitats are distractions.

We agree that the recent paper of Jiang et al. (2020), and in particular the recent paper by Hubau et al (2020 in Nature) (both only published after we submitted our paper) bring into question many projections of the effect of $CO_2$ fertilization using the current generation of land surface models, namely indicating that it is overestimated. Indeed, these recent results could cause no less than a paradigm shift in LSMs and will likely hasten the development of current models. However, it should be noted that the Jiang et al study was on Eucalyptus forest. Also note that until recently, field observations (Lewis et al., 2009, from 1968 and 2007) largely supported our results. Moreover, globally, ORCHIDEE is generally consistent with the global net land sink increase from historical $CO_2$ increase based on FACE experiments -see Liu et al. (2019).

We would propose keeping our full results but adding further discussion and caveats, centred around these new papers.  We feel that while future projections of NPP in response to rising $CO_2$ are highly uncertain in tropical forests where there is no elevated $CO_2$ experiment and where nutrient limitations may decrease $CO_2$ induced enhancements, the increase in the export of carbon to the LOAC and coast (the main/ highlighted result of the future projections) remains an interesting result, and one which is also due to additional drivers such climate change. As such we still feel that these results merit inclusion and discussion, albeit with additional caveats/ discussion reiterating the uncertainty of future projections, and centred around these new publications. We also propose that we change some of the language in this section of the paper (and Abstract) to be more tentative, in reflection these new findings.

Discussion Section 4.1: It is not clear that CO2 enrichment effects on photosynthesis results in enhancement of NPP. Though the comparisons of modeled results with regional estimates of biomass and soil C stocks seem reasonable, the empirical estimates have considerable methodological and sampling uncertainty. L500-502: That the CO2 evasion from the water surfaces is sustained by leaching of dissolved CO2 and DOC from soils is not established. In situ C fixation by wetlands and subsequent decomposition of this material could be a significant source of the CO2 evaded as suggested by Borges, and Abril for the Amazon. Indeed, in lines 530-555, the authors discuss the likely contribution of aquatic macrophytes to the available C, and duly note the difficulty

of incorporating these plants into their model. However, it is therefore odd that this possible contribution is then discounted in lines 555 to 560.

We think that the tentative language used in L500-502 "Our results suggest", in combination with the extensive discussion which you refer to (lines 530-555) appropriately reflect the limitations in our conclusions.

We would debate the conclusion that we have "discounted" the effect of macrophytes or at least that was not our intention. We fully acknowledge the important role that macrophytes are likely to play in sustaining $CO_2$ evasion from the water surface. We only conclude that they are likely to have a limited effect on overall NEP, NBP (only these terms). However, we take the point that the language could be changed to make this clearer and the text will be modified accordingly.

Note also, that that ORCHILEAK represents floodplains as sources of $CO_2$ to the inland water network, from the decomposition of litter and SOC, but also though root respiration of plants in that area. Hence, carbon is not only coming from upland soils, but also from wetland soils and vegetation.

L537-539: It is not correct that strong currents limit the abundance of aquatic macrophytes in the Amazon since most of their growth occurs on floodplains where they can cover large areas.

Ok. This is taken from previous literature (Borges et al., 2015, Scientific Reports) but we can remove this.

L570-572: Both these estimates of the % of NPP per year transferred to inland waters are based on the same model. What are the estimates for the Amazon based on empirical data? L572-582: This discussion of differences between the Amazon and Congo is too simplistic and not representative of the relevant conditions in either system. It would best be deleted unless considerable more information is added. Section 4.2: As noted above, it seems a real stretch to be projecting through the 21st century. L610-625: As this section is written as a comparison with Lauerwald et al. (submitted), it does seem suitable to include until Lauerwald et al is available. Also, there are publications that project hydrological and land use changes in the Amazon.

We will remove the small section comparing the Congo and the Amazon but propose adding a few more limited sentences on the Amazon to the proceeding section (636-645)

L626-624: This paragraph does not seem necessary since these systems are quite different from the Congo and other examples could be selected. Section 4.3: Lines 636-645 re-enforce the issues raised above regarding the projections through the 21st century and the question of whether their inclusion in this paper is warranted.

Conclusion L692-696: Is it likely that an increase in DOC from 9.5 to 11.5 mg C/L will cause ecologically meaningful changes in pH?

It is unclear so we can remove this sentence.

References

Abril, G., Martinez, J.-M., Artigas, L. F., Moreira-Turcq, P., Benedetti, M. F., Vidal, L., … Roland, F. (2013). Amazon River carbon dioxide outgassing fuelled by wetlands. *Nature*, *505*, 395. Retrieved from http://dx.doi.org/10.1038/nature12797

Borges, A. V, Abril, G., Darchambeau, F., Teodoru, C. R., Deborde, J., Vidal, L. O., … Bouillon, S. (2015)b. Divergent biophysical controls of aquatic $CO_2$ and $CH_4$ in the World's two largest rivers. Scientific Reports, 5, 15614. https://doi.org/10.1038/srep15614

Camino-Serrano, M., Guenet, B., Luyssaert, S., Ciais, P., Bastrikov, V., De Vos, B., Gielen, B., Gleixner, G., Jornet-Puig, A., Kaiser, K., Kothawala, D., Lauerwald, R., Peñuelas, J., Schrumpf, M., Vicca, S., Vuichard, N., Walmsley, D., and Janssens, I. A.: ORCHIDEE-SOM: modeling soil organic carbon (SOC) and dissolved organic carbon (DOC) dynamics along vertical soil profiles in Europe, Geosci. Model Dev., 11, 937–957, https://doi.org/10.5194/gmd-11-937-2018, 2018.

Gumbricht, T., Roman-Cuesta, R. M., Verchot, L., Herold, M., Wittmann, F., Householder, E., Murdiyarso, D. (2017). An expert system model for mapping tropical wetlands and peatlands reveals South America as the largest contributor. *Global Change Biology*, *23*(9), 3581–3599. https://doi.org/10.1111/gcb.13689

Heinimann A, Mertz O, Frolking S, Egelund Christensen A, Hurni K, Sedano F, et al. (2017) A global view of shifting cultivation: Recent, current, and future extent. PLoS ONE 12(9): e0184479. https://doi.org/10.1371/journal.pone.0184479

Hubau, W., Lewis, S.L., Phillips, O.L. *et al.* Asynchronous carbon sink saturation in African and Amazonian tropical forests. *Nature* **579,** 80–87 (2020). https://doi.org/10.1038/s41586-020-2035-0

Jiang, M., Medlyn, B.E., Drake, J.E. *et al.* The fate of carbon in a mature forest under carbon dioxide enrichment. *Nature* **580,** 227–231 (2020). https://doi.org/10.1038/s41586-020-2128-9

Krinner, G., Viovy, N., de Noblet-Ducoudré, N., Ogée, J., Polcher, J., Friedlingstein, P., Ciais, P., Sitch, S., and Prentice, I. C. ( 2005), A dynamic global vegetation model for studies of the coupled atmosphere-biosphere system, *Global Biogeochem. Cycles*, 19, GB1015, doi:10.1029/2003GB002199.

Lauerwald, R., Regnier, P., Camino-Serrano, M., Guenet, B., Guimberteau, M., Ducharne, A., … Ciais, P. (2017). ORCHILEAK (revision 3875): a new model branch to simulate carbon transfers along the terrestrial--aquatic continuum of the Amazon basin. *Geoscientific Model Development*, *10*(10), 3821–3859. https://doi.org/10.5194/gmd-10-3821-2017

Lewis, S. L., Lopez-Gonzalez, G., Sonké, B., Affum-Baffoe, K., Baker, T. R., Ojo, L. O., … Wöll, H. (2009). Increasing carbon storage in intact African tropical forests. Nature, 457, 1003. Retrieved from https://doi.org/10.1038/nature07771

Liu, Y., Piao, S., Gasser, T., Ciais, P., Yang, H., Wang, H., … Wang, T. (2019). Field-experiment constraints on the enhancement of the terrestrial carbon sink by $CO_2$ fertilization. Nature Geoscience, 12(10), 809–814. https://doi.org/10.1038/s41561-019-0436-1

Manzoni, S., Taylor, P., Richter, A., Porporato, A. and Ågren, G.I. (2012), Environmental and stoichiometric controls on microbial carbon-use efficiency in soils. New Phytologist, 196: 79-91. doi:10.1111/j.1469-8137.2012.04225.x

Rasera, M. F. F. L., Krusche, A. V., Richey, J. E., Ballester, M. V. R., and Victória, R. L. (2013). Spatial and temporal variability of pCO2 and CO2 efflux in seven Amazonian Rivers. *Biogeochemistry*, *116*(1), 241–259. https://doi.org/10.1007/s10533-013-9854-0

Richey, J. E., Melack, J. M., Aufdenkampe, A. K., Ballester, V. M., & Hess, L. L. (2002). Outgassing from Amazonian rivers and wetlands as a large tropical source of atmospheric CO2. *Nature*, *416*, 617. Retrieved from http://dx.doi.org/10.1038/416617a

Tyukavina, A., Hansen, M. C., Potapov, P., Parker, D., Okpa, C., Stehman, S. V, … Turubanova, S. (2018). Congo Basin forest loss dominated by increasing smallholder clearing. Science Advances, 4(11). https://doi.org/10.1126/sciadv.aat2993

---

## Author Response (AR1)

**Response to reviewer 1 (response in red).**

We thank Reviewer 1 for taking the time to review the paper. We agree that additional validation, of pCO2 would improve the paper, as well as some additional analysis of the causes for the discrepancy between our estimates of riverine FCO2 vs those of Borges et al (2019), such as gas transfer velocity and stream surface area. We respond in more detail (point by point) below (excerpts from the text are in quotation marks-in some longer experts we have highlighted particularly important sentences in bold).

The model validation step is very slim, as authors compare the model outputs of dissolved organic carbon (DOC) to a sub-set of published field data. DOC in tropical rivers is extremely refractory and provides little grasp on aquatic carbon cycling as the most labile DOC fraction is very rapidly mineralized (both in soils and in water). So DOC provides a poor validation of the carbon cycling in the river, and might be considered almost as a passive tracer and provide a rough validation of the hydrological connectivity between soils and rivers. Conversely, a convincing validation of the model would be to compare the model outputs of the dissolved $CO_2$ concentration (or the corresponding partial pressure of $CO_2$) with the extensive field data collected by Borges et al. (2019) that are publically available (Borges and Bouillon 2019). While a point by point comparison would not make sense, it could be useful to check if the model captures the overall range of spatial variations along the river and among the different stream sizes. Such validation would be extremely convincing because the $CO_2$ evasion from the river to the atmosphere (the core topic of the paper) is computed from the dissolved $CO_2$ concentration (the atmospheric $CO_2$ is comparatively invariant) and the gas transfer velocity. So if the model does not represent correctly the dissolved $CO_2$ concentration then this implies that the $CO_2$ evasion rates are incorrect (as well as any conclusion based on past reconstruction and future projection from the model outputs).

Thank you for these suggestions. We have conducted some analysis (in line with your suggestions-including against observed pCO2) and added additional text.

Please see lines 387-414:

"In Figure 5, we compare simulated DOC concentrations at six locations (Fig. 1) along the Congo River and Oubangui tributary, against the observations of Borges at al. (2015[b]). We show that we can recreate the spatial variation in DOC concentration within the Congo basin relatively closely with an $R^2$ of 0.74 and an RMSE of 23% (Fig. 5). We are also able to simulate the broad spatial pattern of $p$CO$_2$ measured in Borges et al. (2019). During high flow season (mean of 6 consecutive months of highest flow, 2009-2019) we simulate a mean $p$CO$_2$ of 3,373 ppm and 5,095 ppm at Kisangani and Kinshasa (Brazzaville) respectively, compared to the observed values of 2,424 ppm and 5,343 ppm during high water (measured in December 2013, Borges et al., 2019) (Table 3). Similarly, during low flow season (mean of 6 consecutive months of lowest flow, 2009-2019) we simulate a mean $p$CO$_2$ of 1,563 ppm and 2,782 ppm at Kisangani and Kinshasa respectively, compared to the observed values of 1,670 ppm and 2,896 ppm during falling water (June 2014, Borges et al., 2019) (Table 3).

While we are able to recreate observed spatial differences in DOC and $p$CO$_2$, as well as broad seasonal variations, we are not able to correctly predict the exact timing of the simulated highs and lows, a reflection of not fully capturing the hydrological seasonality. For example, our mean June $p$CO$_2$ at Kinshasa (Brazzaville) is 4,470 ppm, while Borges et al measured a mean of 2,896 ppm (Table

3). However, our value for July of 2,621 ppm is much closer, and moreover our mean value for December of 5,154 ppm is relatively close to the observed value of 5,343 ppm. Similarly, we fail to predict the timing of the June falling water at Kisangani (Table 3).

In Figure 6, we compare simulated $pCO_2$ against the observed monthly time series at Bangui on the Oubangui River (Bouillon et al., 2012 & 2014), as far as we are aware the most complete time series of $pCO_2$ published from the Congo basin, spanning March 2010 to March 2012 (with only the single month of June 2010 missing). Again, while the model fails to correctly predict the precise timing of the peak as with the Kinshasa and Kisangani datasets the broad seasonal variation in $pCO_2$ is captured, with the observed and modelled times series ranging from 227- 4040 ppm and 415- 2928 ppm, respectively (Fig. 6)."

We have also added an additional Table (Table 3, see below) and Figure (Fig. 6, see below)

Table 3: Observed (Borges et al., 2019) and modelled $pCO_2$ (in ppm) at Kinshasa (Brazzaville) and Kisangani on the Congo river at various water levels.

| Location | Observed $pCO_2$ highwater (December 2013) | Modelled $pCO_2$ highwater (December Mean 2009-2019) | Modelled $pCO_2$ high flow season (mean of 6 consecutive months of highest flow 2009-2019) | Observed $pCO_2$ falling water (June 2014) | Modelled $pCO_2$ falling water (June mean 2009-2019) | Modelled $pCO_2$ low flow season (mean of 6 consecutive months of lowest flow 2009-2019) |
|---|---|---|---|---|---|---|
| Kinshasa (Brazzaville) | 5,343 | 5,154 | 5,095 | 2,896 | 4,470 | 2,782 |
| Kisangani | 2,424 | 2,166 | 3,373 | 1,670 | 3,126 | 1,563 |

[Figure]

Figure 6: Time series of observed *versus* simulated $pCO_2$ at Bangui on the River Oubangui. Observed data is from Bouillon et al., 2012 and Bouillon et al., 2014.

The overall emission of CO2 from the fluvial component of the Congo Basin (TgC/yr) is based on the product of a CO2 flux density (mol/m2/yr) and a stream surface area; the areal CO2 flux is itself computed from the air-water CO2 concentration gradient, and the gas transfer velocity; the air-water CO2 concentration gradient in turn is mainly function of the dissolved CO2 concentration. So there are three quantities that could explain the difference between the 4 estimates of integrated CO2emissions discussed in section L-513-517: the CO2 dissolved concentration, the gas transfer velocity and the stream surface area. The evaluation of the model performance would be much more convincing if each of these three quantities was compared to available estimates.

In response to these comments we have added additional discussion (now lines 572-641):

[revised manuscript text omitted]

Final elements of validation and discussion that are missing are the contribution of HCO3- to the export of dissolved inorganic carbon (DIC) from the river to the ocean. The export of DIC from rivers to the ocean is mainly in the form of HCO3- including the Congo river (Wang et al. 2013). So ignoring HCO3- would lead to a very substantial under-estimation of DIC export to the ocean. This should be fairly easy to implement with a weathering model and GIS of lithology. This is of course of interest for the topic of paper as weathering intensity is a function of temperature and precipitation that are used by the authors to study the long-term (1861-2099) trends of aquatic carbon fluxes. Further, there are substantial data-sets of total alkalinity (that mainly corresponds to HCO3-) providing spatial (Borges et al. 2019) and seasonal (Wang et al. 2013; Bouillon et al. 2012; 2013) patterns of HCO3- variability. Additionally, dissolved CO2 is in thermodynamic equilibrium with HCO3-, so it is required to have some grasp on HCO3- variability to correctly model dissolved CO2 dynamics, hence, CO2 emissions to the atmosphere.

We appreciate these comments and suggestions and acknowledge that the lack of accounting for HCO3- is a limitation of the model. However, we would consider creating and implementing a weathering model outside the scope of our study and therefore as an alternative, we have added estimates from the literature and discuss these limitations.

Please see lines 642-658:

"Our simulated export of C to the coast of 15 (15.3) Tg C yr$^{-1}$ is virtually identical to the TOC+DIC export estimated by Borges et al. (2015[a]) of 15.5 Tg C yr$^{-1}$, which is consistent with the fact that we simulate a similar spatial variation of DOC concentrations (Fig. 8 and Fig. 1 for locations). It is also relatively similar to the 19 Tg C yr$^{-1}$ (DOC + DIC) estimated by Valentini et al. (2014) in their synthesis of the African carbon budget. Valentini et al. (2014) used the largely empirical based Global Nutrient Export from WaterSheds (NEWS) model framework and they point out that Africa was underrepresented in the training data used to develop the regression relationships which underpin the model, and thus this could explain the small disagreement.

Of the total 15 Tg C yr$^{-1}$ exported to the coast, we simulate a 2.4 Tg C yr$^{-1}$ component of dissolved CO$_2$, which is relatively similar to the empirically derived estimate of the total DIC export of 3.3 Tg C yr$^{-1}$ calculated in Wang et al. (2013). According to Wang et al., dissolved CO$_2$ accounts for the majority (1.9 Tg C yr$^{-1}$) with the rest being the weathering derived flux of HCO$_3^-$. Thus, the discrepancy between the two estimates is likely to be largely caused by our lack of accounting for the weathering derived flux (HCO$_3^-$) which they estimate at 1.4 Tg C yr$^{-1}$. In summary, despite this model limitation the results of Wang et al. (2013) suggest that we still capture the majority of the DIC flux."

L44 : In this section it's unclear what is meant by "increase" of "net primary productivity" and "storage in tree biomass". A recent study shows that African forests are sinks of carbon on yearly basis, and that the carbon sink is constant in time from the mid-1990's to present (Hubau et al. 2020). So, according to this study, there is no "increase" in NPP as stated but a constant sink. Please clarify.

In response to these comments, we have added additional lines in the introduction (now lines 64-66):

"Moreover, recent field data suggests that the above ground C sink in tropical Africa was relatively stable from 1985 to 2015 (Hubau et al., 2020)."

Please also see additional discussion (now lines 673-698):

"Up to a point, our results also concur with estimates based on the upscaling of biomass observations (Lewis et al., 2009; Hubau et al., 2019). Lewis et al. (2009) up-scaled forest plot measurements to calculate that intact tropical African forests represented a net uptake of approximately 300 Tg C yr$^{-1}$ between 1968 and 2007 and this is consistent with our NEP estimate of 275 Tg C yr$^{-1}$ over the same period. However, more recently an analysis based on an extension of the same dataset found that the above ground C sink in tropical Africa has been relatively stable from 1985 to 2015 (Hubau et al., 2020).

A major source of the uncertainty associated with future projections of NPP and NEP comes from our limited understanding and representation of the $CO_2$ fertilization effect. Recent analysis of data from some of the longest-running Free-Air $CO_2$ Enrichment (FACE) sites, consisting of early-successional temperate ecosystems, found a 29.1 ± 11.7% stimulation of biomass over a decade (Walker et al., 2019). A meta-analysis (Liu et al., 2019) of seven temperate FACE experiments combined with process-based modelling also found substantial sensitivity (0.64 ± 0.28 PgC yr$^{-1}$ per hundred ppm) of biomass accumulation to atmospheric $CO_2$ increase, and the same study showed that ORCHIDEE model simulations were largely consistent with the experiments. However, other FACE experiments on mature temperate forests (Körner et al., 2005), as well as eucalyptus forests bring into question whether the fertilization effects observed in temperate FACE experiments can be extrapolated to other ecosystems. For example, the Swiss FACE study, a deciduous mature forest, found no significant biomass increase with enhanced $CO_2$ (Körner et al., 2005), while a FACE experiment on a mature eucalyptus forest in Australia found that while $CO_2$ stimulated an increase in C uptake through GPP, this did not carry to the ecosystem level, largely as a result of a concurrent increase in soil respiration (Jiang et al., 2020). Unfortunately, no results are yet available from any tropical FACE experiments, though the Amazon FACE experiment is underway and the eventual results will be crucial in developing our understanding of the $CO_2$ fertilization effect beyond the temperate zone."

L74 : The tropical region is also a hotspot of aquatic C cycling due to wetland productivity and wetland carbon inputs to rivers (Abril & Borges 2019), in addition to "terrestrial NPP". In the Amazon river a large fraction of fluvial CO2 emissions to the atmosphere are sustained by wetland inputs (Abril et al. 2014). L74: The fact that the "tropical region is a hotspot area for inland water C cycling" (as stated) was authoritatively demonstrated by seminal papers such as Richey et al. (2002) and Melack et al. (2004), more than a decade before the recent Lauerwald et al. (2015) work. L 78-80: there are (at least) two additional elements of context that could be relevant for the introduction: 1) the Raymond et al. (2013) and the Lauerwald et al. (2015) estimates are in fact based on the same initial data-base of pCO2 computed from pH and alkalinity (GLORICH) that was extrapolated globally using two different approaches; this illustrates how uncertain these global estimates are, since the resulting values differ by a factor of 3; 2) While the conclusion of Lauerwald et al. (2015) that the majority of CO2 emissions from rivers comes from the tropics is probably correct, field data (used in the global extrapolation) is nearly absent in the tropics. For instance for the Congo River there is only one single data entry in the GLORICH data-set. Most of the data used to develop the statistical model of Lauerwald et al. (2015) come from non-tropical areas such as North America and Scandinavia.

We have changed the references to prioritise empirical estimates and also added additional context for the Lauerwald et al. (2015) and Raymond et al. (2013) estimates.

Please see lines 81-94:

"The tropical region is a hotspot area for inland water C cycling (Richey et al., 2002; Melack et al., 2004; Abril et al., 2014; Borges et al., 2015[a]; Lauerwald et al., 2015) due to high terrestrial NPP and precipitation, and a recent study used an upscaling approach based on observations to estimate present day $CO_2$ evasion from the rivers of the Congo basin at 251±46 Tg C $yr^{-1}$ and the lateral C (TOC +DIC) export to the coast at 15.5 (13-18) Tg C $yr^{-1}$ (Borges at al., 2015[a]; Borges et al., 2019). To put this into context, their estimate of aquatic $CO_2$ evasion represents 39% of the global value estimated by Lauerwald et al. (2015, 650 Tg C $yr^{-1}$) or 14% of the global estimate of Raymond et al. (2013, 1,800 Tg C $yr^{-1}$). **Note that while Lauerwald et al. (2015) and Raymond et al. (2013) relied largely on the same database of $pCO_2$ measurements (GloRiCh, Hartmann et al., 2014) as the basis for their estimates, they took different, albeit both empirically led approaches. Moreover, both approaches were limited by a relative paucity of data from the tropics, which also explains the high degree of uncertainty associated with our understanding of global riverine $CO_2$ evasion.**"

L 244: Borges et al. (2019) report discharge data from the mainstem Congo at Kisangani. So there are additional data-sets to validate the model hydrology.

We performed the Hydrology validation before this paper was published. Moreover, we were unable to access this dataset.

L513: It could be useful to put into context how these different fluxes were computed. The Raymond et al. (2013) estimate is based on a single pCO2 value (apparently from pH and alkalinity measurements in Pool Malebo) that was extrapolated to the whole basin. The comparison of this single value of pCO2 with the extensive data set reported by Borges et al. (2019) shows that it is unrealistically low (refer to Supplemental Figure 18). Lauerwald et al. (2015) et al. estimate of pCO2 compares better to the Borges et al. (2019) data-set but still fails to represent the influence of the Cuvette Centrale (refer to Supplemental Figure 18). Please also note that the CO2 estimate for the Congo reported by Borges et al. (2015) was based on exactly the same stream surface area and gas transfer velocity as those used by Raymond et al. (2013), and also showed that the Raymond et al. (2013) estimate was under-estimated (obviously, since the pCO2 value is unrealistically low). So there is some clear convergence that the present estimate of CO2 emission based on ORCHILEAK is under-estimated even if it is coincidentally close to the one reported by Raymond et al. (2013). The actual reasons of the under-estimation need to be explored as suggested in the above Major Comments.

We have added some additional explanation (now lines 572-582):

"Our estimate of the integrated present-day aquatic $CO_2$ evasion from the river surface of the Congo basin (32 Tg C $yr^{-1}$) is the same as that estimated by Raymond et al. (2013) (also 32 Tg C $yr^{-1}$), downscaled over the same basin area, but smaller than the 59.7 Tg C $yr^{-1}$ calculated by Lauerwald et al. (2015) and far smaller than that of Borges et al. (2015[a]), 133-177 Tg C $yr^{-1}$ or Borges et al. (2019), 251±46 Tg C $yr^{-1}$. **The recent study of Borges et al. (2019) is based on by far and away the most extensive dataset of Congo basin $p$CO$_2$ measurements to date and thus suggests that we substantially underestimate total riverine CO$_2$ evasion**. As previously discussed, we simulate the broad spatial and temporal variation in observed DOC and $p$CO$_2$ (2015[a, b], Fig. 5, Table 3) relatively well. It is therefore somewhat surprising that our basin-wide estimate of riverine CO$_2$ evasion is so different. Below we discuss some possible explanations for this discrepancy related to methodological differences and limitations."

Please also see the proceeding paragraphs for further discussion of the potential causes for our underestimation of the aquatic Co2 evasion from the river surface.

L124: Why is the water surface area varied diurnally?

This is the time step for the routing scheme of ORCHILEAK and the surface area varies with discharge.

Figure 1. The figure needs latitudes and longitudes indicated. Lake Tanganyika is drawn as if a loop of rivers; redraw as a lake.

This figure has been changed accordingly (line 165):

[Figure]

**Figure 1:**Extent of the Congo Basin, central quadrant of the "Cuvette Centrale" and sampling
stations (for DOC and discharge) along the Congo and Oubangui Rivers (in italic).

Figure 2 and associated text (L153-168) do not consider the veracity of these data. Though 13 plant functional groups (pft) are prescribed, how well are their ecophysiological characteristics in the conditions of the Congo known? 'Tropical broadleaved raingreen trees' is an odd phrase. Section 2.2: Given the importance of the wetlands to the modeling, further discussion of datasets used is warranted.

We take these points. While it is difficult to find directly comparable empirical estimates for the Congo Basin for which to compare the PFTs against, we have added a few sentences detailing a broader comparison (now lines 177-185):

"Most published estimates for land-cover follow national boundaries and so we can make broad comparisons with published estimates for the Democratic Republic of Congo (DRC). For example, our value for total forest cover for the DRC (65%), is close to the 67% and 68% values estimated by the Congo Basin Forest Partnership (CBFP, 2009), and Potapov et al. (2012), respectively. Agriculture covers only a small proportion of the basin according to the LUH dataset that is based on FAO cropland area statistics, with C3 (PFT12, Fig. 2 g) and $C_4$ (PFT13, Fig. 2 h) agriculture making up a maximum basin area of 0.5 and 2% respectively. In reality, a larger fraction of the basin is composed of small scale and rotational agriculture (Tyukavina et al., 2018)."

L177: What is the definition of swamps versus floodplains and how are they distinguished in the Congo?

In terms of determining the maximum extent of swamps and floodplains forcing files, these are taken from the dataset of Gumbricht et al. (2017) and therefore follow his definitions.  His definition of swamps can be found in Table 1 of his paper but the main characteristics are as follows: "Usually bound to valleys and plains; planar surfaces. Wet all year around, but not necessarily inundated. Usually tree covered."

The max floodplain is defined by aggregating all of the wetland categories in the Gumbricht dataset (including swamps).

In terms of the representation in ORCHILEAK, the difference between swamps and floodplains is outlined in section 2.2. See Lauerwald et al. (2017) for further details.

In response to these comments we have also modified the corresponding paragraph (now lines 199-217):

"In grids where swamps exist, a constant proportion of river discharge is fed into the base of the soil column; **ORCHILEAK does not explicitly represent a groundwater reservoir and so this imitates the hydrological coupling of swamps and rivers through the groundwater table.** The maximal proportions of each grid which can be covered by floodplains and swamps are prescribed by the maximal fraction of floodplains (MFF) and the maximal fraction of swamps (MFS) forcing files respectively (Guimberteau et al., 2012). See also Lauerwald et al. (2017) and Hastie et al. (2019) for further details. We created an MFF forcing file for the Congo basin, derived from the Global Wetlands[v3] database; the 232 m resolution tropical wetland map of Gumbricht et al. (2017) (Fig. 3 a and b). We firstly amalgamated all the categories of wetland (which include floodplains and swamps) before aggregating them to a resolution of 0.5° (the resolution at which the floodplain/swamp forcing files are read by ORCHILEAK), assuming that this represents the maximum extent of inundation in the basin. This results in a mean MFF of 10%, i.e. a maximum of 10% of the surface area of the Congo basin can be inundated with water. This is identical to the mean MFF value of 10% produced with the Global Lakes and Wetlands Database, GLWD (Lehner, & Döll, P.,2004; Borges et al., 2015[b]). We also created an MFS forcing file from the same dataset (Fig. 3 c and d), merging the 'swamps' and 'fens' wetland categories (although note that there are virtually no fens in the Congo basin) from Global Wetlands[v3] database (Gumbricht et al., 2017) and again aggregating them to a 0.5° resolution. Please see Table 1 of Gumbricht et al. (2017) for further details."

L178: Does inundation of the floodplains require exceedance of 'bank-full discharge'? See comment about section 2.3.

Yes, it does, and bank-full discharge is defined as the median stream flow over the period 1990 to 2005.

In response to these comments we have also modified the corresponding text to make it clearer (now lines- 255-258):

"As in previous studies on the Amazon basin (Lauerwald et al. 2017, Hastie et al., 2019) we defined bank-full discharge, i.e. the threshold discharge at which floodplain inundation starts (i.e. overtopping of banks), as the median discharge (50th percentile i.e. $streamr_{50th}$) of the present-day climate forcing period (1990 to 2005)."

L179-180: It is unclear why 'a constant proportion of river discharge is fed into the base of the soil column'.

ORCHILEAK does not yet explicitly represent a groundwater reservoir. This imitates how rivers and swamps are hydrologically coupled through the groundwater table.

In response to these comments we have also modified the corresponding text to make it clearer (now lines- 199-201):

"In grids where swamps exist, a constant proportion of river discharge is fed into the base of the soil column; ORCHILEAK does not explicitly represent a groundwater reservoir and so this imitates the hydrological coupling of swamps and rivers through the groundwater table."

Please see section 2.1.2 and Figure 3 of Lauerwald et al. (2017) for a more detailed explanation.

L188-190: Round the MFF to 10%. Is this value the maximum MFF or the mean maximum?

Done (rounded to 10%)

Mean maximum

L193: How are 'fens' different from swamps in the Congo?

We have merged the swamps and fens categories from Gumbricht et al. (2017) so effectively they are not different in our study. Irrespectively, according to the Gumbicht dataset there are virtually no fens in the Congo.

See modified lines 213-217:

"We also created an MFS forcing file from the same dataset (Fig. 3 c and d), merging the 'swamps' and 'fens' wetland categories (although note that there are virtually no fens in the Congo basin) from Global Wetlands[v3] database (Gumbricht et al., 2017) and again aggregating them to a 0.5° resolution. Please see Table 1 of Gumbricht et al. (2017) for further details."

Section 2.3: Indeed, simulating the hydrology well is critical. The description of the calibration steps is somewhat confusing. For example, line 217 states 'Without calibration, the majority of the different climate forcing model runs performed poorly .'. However, key hydrological parameters needed calibration. Hence, it would seem issues with both forcings and model parameters are confounded.

Virtually all hydrological models require calibration through the modification of model parameters. Admittedly, the forcing datasets generally do not perform as well for the Congo as for the Amazon for example (likely a result of more climate data being available for the gridded climate forcing fields in the Amazon), which is why we tested several different climate forcing data to estimate uncertainties. The performance without calibration would not have been acceptable/ reasonable, and we feel that we stuck the right balance between improving river flow simulation and over-calibration of parameters, keeping in mind the limitations of climate forcing datasets, and that we are calibrating/ validating for 3 quite different situations (the main stem of the Congo, a much smaller tributary, and overall inundation area).

L233-240: The concept of bank-full discharge as a threshold for initiation of inundation of floodplains is questionable as applied to tropical floodplain such as those in the Amazon or Congo. Studies inundation dynamics in the Amazon with detailed measurements or modeling indicate that inundation occurs more or less continuously as the rivers rise and that the water comes from both the rivers and uplands (e.g., Lesack and Melack 1995 Water Resources Res 31:329–334; Bonnet et al. 2017 Hydrol. Processes 31: 1702–1718; Rudorff et al 2014 Water Resources Res 31:329–349; Ji et al. 2019 Water Resources Res 54).

While we appreciate this point, we would respectfully debate some of your conclusions. For example, in their paper Bonnet et al. (2017) conclude that "The mainstream was the main input of water to the flooded area, accounting on average for 93% of total water inputs by the end of the water year. Direct precipitation and runoff from uplands contributed less than or equal to 5% and 10%, respectively. The seepage contribution was less than 1%". They go on to explain that in their model "Diffusive overbank flows occur where the mainstream water level is above levee crests."

Similarly, Rudorff et al. (2014) conclude that "Diffuse overbank flows represent 93% of total river to floodplain discharge"

It is true that Lesack and Melack (1995) find a much higher percentage of inflow coming from runoff (57%) but this is the results from a single case study of a small lake in the central Amazon basin.

Moreover, the majority of the wetlands which we represent in the Congo in ORCHILEAK are swamps, and so do not rely on overtopping at bank-full discharge.

L248-249: The algorithms used to generate the GIEMS vary in their effectiveness depending the density and extent of the inundated vegetation. Section 2.4.1: How well do the soil processes derived for Europe (Camino Serrano et al. 2018) apply to the Congo, how were the passive, slow and active pools determined and how were the decomposition rates in the flooded and non-flooded soil derived?

One of the main limitations which Camino Serrano et al. (2018) identified with the potential application of ORCHIDEE-SOM to the tropics was the lack of representation of DOC coming from throughfall, which is incorporated into ORCHILEAK (Lauerwald et al., 2017). The other main limitation in applying it to the Congo is the lack of an explicit representation of peatlands which we discuss in detail (see lines 735 to 760).

The active passive and slow pools are explained in detail on page 3832 of Lauerwald et al. (2017) but the main part of the text is as follows " The soil carbon module distinguishes 3 different pools of DOC depending on the source material: active, slow and passive (Camino Serrano et al, 2018 - GMD). The DOC derived from the active SOC pool and metabolic litter is assigned to the active DOC pool, while the DOC derived from the slow and passive SOC pools are assigned to the slow and passive DOC pools, respectively (Eqs. 43–45). A part of DOC derived from structural plant litter, which is related to the lignin structure of the litter pool (Krinner et al., 2005), is allocated to the slow DOC pool, while the remainder feeds the active DOC pool. The proportion of the decomposed litter and SOC that is transformed into DOC instead of $CO_2$ depends on the carbon use efficiency (CUE), set here to a value of 0.5 (Manzoni et al., 2012). Taken that the same residence time for the slow and passive DOC pools is used in ORCHIDEE-SOM (Camino Serrano, 2015), we merge these two pools when computing throughfall and lateral transport of DOC. Thus, the labile pool is identical to the active pool of the soil carbon module, while the refractory pool combines the slow and passive pools. The labile ($FTF,DOClab$ ) and refractory ($FTF,DOCref$ ) proportions of throughfall DOC are added to the active and slow DOC pools of the first soil layer, respectively" We acknowledge the fact that these modelled SOC pools are not measurable, as in any land surface model, and there is no sufficient radiocarbon age data in Congo to accurately calibrate SOC turnovers in the model.

Moreover, note that in ORCHILEAK decomposition rates of SOC, DOC and litter in flooded soils are 3x lower that of those in non- flooded soils. This is based on the findings by Rueda-Delgado et al., (2006) but also supported by additional research such as Dos Santos & Nelson., (2013).

Section 2.4.2: What were the projected land use changes? These would seem rather difficult to prescribe, as noted in the text. The exclusion of shifting cultivation would seem a serious omission.

The main land-use changes are detailed in Figure A2 of the Appendix. We acknowledge the fact that exclusion of shifting cultivation is a major limitation, though one which would be difficult to incorporate in view of the lack of a spatially explicit dataset. The LUH1 reconstruction indicates for instance shifting cultivation affecting all the tropics with a residence time of agriculture of 15 years, whereas the review from Heininan et al. 2017 (Plos one) revised downwards the area of this type of agriculture, with generally low values in Congo, except in the North east and South East, but suggested a shorter turnover of agriculture of two years only. In view of such uncertainties, we did not include shifting agriculture in the model. But added in the discussion the possibility to improve this situation using new remote sensing datasets on high resolution land cover change (Tyukavina et al. 2018, Sci. Adv)

In response to these comments we have also added to the corresponding text.

Lines 304-311:

"In the paper which describes the development of the future land use change scenarios under RCP 6.0 (Hurtt et al., 2011), it is shown that land use change is highly sensitive to land use model assumptions, such as whether or not shifting cultivation is included. The LUH1 reconstruction for instance indicates shifting cultivation affecting all of the tropics with a residence time of agriculture of 15 years, whereas the review from Heinimann et al. (2017) revised downwards the area of this type of agriculture, with generally low values in Congo, except in the North East and South East, but suggested a shorter turnover of agriculture of two years only. In view of such uncertainties, we did not include shifting agriculture in the model."

Lines 769-771:

"Finally, the issue of shifting cultivation demands further attention; at least for the present day a shifting cultivation forcing file could be developed based on remote sensing data (Tyukavina et al., 2018)."

Results Section 3.1: In general, simulations of mean monthly discharge for large tropical river systems without large dams at downstream stations has been demonstrated as feasible with several models. Hydrological simulations can become increasingly difficult as the scale decreases, as indicated by the less successful simulations of the Ubangi River. Though the text comparing the GIEMS and simulated inundated areas makes sense, the issue of topography as a factor influencing simulated inundated area deserves mention. L358-362: These judgments should be left to the reader to make.

We accept these points and have removed these sentences accordingly.

Section 3.2: What is the basis for the calculated standard deviations for the fluxes? Figure 5 would be clearer if redrafted larger with simpler graphics. Given all the uncertainties in the modeling and underlying data, Figure 6 would seem quite questionable.

The standard deviation represents the interannual variation across the relevant period (for example 1981-2010.  We have made Figure 5 (now Figure 7) larger and simpler (note that a high definition version will be submitted at a later stage). While we note and indeed discuss the uncertainties in detail, we feel that Fig 6 (now Fig. 8) is still interesting and illustrates the fact that the Cuvette Centrale is a hotspot region of exchange between the terrestrial and aquatic realms.

Section 3.3: These results seem premature without a thorough, rigorous evaluation of the model's output under current conditions. Section 3.4: 'The dramatic increase in the concentration of atmospheric CO2 (Fig. 8 g) and subsequent fertilization effect on terrestrial NPP has the greatest overall impact on all of the fluxes across the simulation period' is a critical point and raises a fundamental question about the veracity of the projected changes. As illustrated in a recent paper (Jiang et al. 2020 Nature 580:227-231), the possible CO2 enrichment effects on mature forests are not well captured by current models and need considerably more work to be understood and properly incorporated into models. Figure 9 would be clearer if redrafted larger with simpler graphics. The colors and simple depictions of habitats are distractions.

In response to these comments we have added substantial additional discussion.

Lines 661-698:

"There is relatively sparse observed data available on the long-term trends of terrestrial C fluxes in the Congo.  Yin et al. (2017) used MODIS data to estimate NPP between 2001 and 2013 across central Africa. They found that NPP increased on average by 10 g C m$^{-2}$ per year, while we simulate an average annual increase of 4 g C $m^{-2}$ $yr^{-1}$ over the same period across the Congo Basin. The two values are not directly comparable as they do not cover precisely the same geographic area but it is encouraging that our simulations exhibit a similar trend to remote sensing data. As previously noted, MODIS derived estimates of NPP do not fully include the effect of $CO_2$ fertilization (de Kauwe et al., 2016) whereas ORCHILEAK does. Thus, the MODIS NPP product may underestimate the increasing trend in NPP, which would bring our modeled trend further away from this dataset. On the other hand, forest degradation effects and recent droughts have been associated with a decrease of greenness (Zhou et al., 2014) and above ground biomass loss (Qie et al., 2019) in tropical forests.

Up to a point, our results also concur with estimates based on the upscaling of biomass observations (Lewis et al., 2009; Hubau et al., 2019). Lewis et al. (2009) up-scaled forest plot measurements to calculate that intact tropical African forests represented a net uptake of approximately 300 Tg C $yr^{-1}$ between 1968 and 2007 and this is consistent with our NEP estimate of 275 Tg C $yr^{-1}$ over the same period. However, more recently an analysis based on an extension of the same dataset found that the above ground C sink in tropical Africa has been relatively stable for the three decades leading to 2015 (Hubau et al., 2020).

A major source of the uncertainty associated with future projections of NPP and NEP comes from our limited understanding and representation of the $CO_2$ fertilization effect. Recent analysis of data from some of the longest-running Free-Air $CO_2$ Enrichment (FACE) sites, consisting of early-successional temperate ecosystems, found a 29.1 ± 11.7% stimulation of biomass over a decade (Walker et al., 2019). A meta-analysis (Liu et al., 2019) of seven temperate FACE experiments combined with process-based modelling also found substantial sensitivity (0.64 ± 0.28 PgC $yr^{-1}$ per hundred ppm) of biomass accumulation to atmospheric $CO_2$ increase, and the same study showed that ORCHIDEE model simulations were largely consistent with the experiments. However, other FACE experiments on mature temperate forests (Körner et al., 2005), as well as eucalyptus forests bring into question whether the fertilization effects observed in temperate FACE experiments can be extrapolated to other ecosystems. For example, the Swiss FACE study, a deciduous mature forest, found no significant biomass increase with enhanced $CO_2$ (Körner et al., 2005), while a FACE experiment on a mature eucalyptus forest in Australia found that while $CO_2$ stimulated an increase in C uptake through GPP, this did not carry to the ecosystem level, largely as a result of a concurrent increase in soil respiration (Jiang et al., 2020). Unfortunately, no results are yet available from any tropical FACE experiments, though the Amazon FACE experiment is underway and the eventual results will be crucial in developing our understanding of the $CO_2$ fertilization effect beyond the temperate zone."

Lines 728-732:

"There are also considerable uncertainties associated with future climate projections in the Congo basin (Haensler et al., 2013). Nutrient limitation on growth and a better representation of effect of enhanced $CO_2$, particularly with regards to soil respiration (Jiang et al., 2020) and tree mortality (Hubau et al., 2020), are two crucial aspects which need to be further developed."

We have also made wording more tentative throughout (see Abstract for example and conclusion for example).

Figure 9 (now figure 10) has been simplified. As has the corresponding figure for the present day (now figure 7).

[Figure]

**Figure 10:** Annual C budget (NBP) for the Congo basin for; left, the Year 1861 and right, the
Year 2099, simulated with ORCHILEAK. NPP is terrestrial net primary productivity, TF is
throughfall, SHR is soil heterotrophic respiration, $FCO_2$ is aquatic CO2 evasion, LOAC is C
leakage to the land-ocean aquatic continuum ($FCO_2 + LE_{Aquatic}$), LUC is flux from Land-use
change, and $LE_{Aquatic}$ is the export C flux to the coast. Range represents the standard deviation

Discussion Section 4.1: It is not clear that CO2 enrichment effects on photosynthesis results in enhancement of NPP. Though the comparisons of modeled results with regional estimates of biomass and soil C stocks seem reasonable, the empirical estimates have considerable methodological and sampling uncertainty. L500-502: That the CO2 evasion from the water surfaces is sustained by leaching of dissolved CO2 and DOC from soils is not established. In situ C fixation by wetlands and subsequent decomposition of this material could be a significant source of the CO2 evaded as suggested by Borges, and Abril for the Amazon. Indeed, in lines 530-555, the authors discuss the likely contribution of aquatic macrophytes to the available C, and duly note the difficulty of incorporating these plants into their model. However, it is therefore odd that this possible contribution is then discounted in lines 555 to 560.

We think that the tentative language used (now lines L559-561) "Our results suggest", in combination with the extensive discussion which you refer to appropriately reflect the limitations in our conclusions.

We would debate the conclusion that we have "discounted" the effect of macrophytes or at least that was not our intention. We fully acknowledge the important role that macrophytes are likely to play in sustaining $CO_2$ evasion from the water surface. However, we take the point that the language could be changed to make this clearer and have modified the text accordingly.

Note also, that that ORCHILEAK represents floodplains as sources of $CO_2$ to the inland water network, from the decomposition of litter and SOC, but also though root respiration of plants in that area. Hence, carbon is not only coming from upland soils, but also from wetland soils and vegetation.

Please see modified paragraph (now lines 595-625):

"The difference in our simulated riverine $CO_2$ evasion compared to the empirically derived estimate of Borges et al. (2019), could be caused by the lack of representation of aquatic plants in the ORCHILEAK model. Borges et al. (2019) used the stable isotope composition of $\delta^{13}$C-DIC to determine the origin of dissolved $CO_2$ in the Congo River system and found that the values were consistent with a DIC input from the degradation of organic matter, in particular from $C_4$ plants. Crucially, they further found that the $\delta^{13}$C-DIC values were unrelated to the contribution of *terra-firme* $C_4$ plants, rather that they were more consistent with the degradation of aquatic $C_4$ plants, namely macrophytes. ORCHILEAK does not represent aquatic plants, and the wider LSM ORCHIDEE does not have an aquatic macrophyte PFT either (though root respiration of floodplain plants for the PFTs represented, is accounted for as a C source). This could at the very least partly explain our conservative estimate of river $CO_2$ evasion, given that tropical macrophytes have relatively elevated NPPs. Rates as high as 3,500 g C m$^{-2}$ yr$^{-1}$ have been measured on floodplains in the Amazon (Silva et al., 2009). While this value is higher than the values simulated in the Cuvette Centrale by ORCHILEAK (Figure 8), they are of the same order of magnitude and so this alone cannot fully explain the discrepancy compared to the results of Borges et al. (2019). In the Amazon basin it has been shown that wetlands export approximately half of their gross primary production (GPP) to the river network compared to upland (*terra-firme*) ecosystems which only export a few percent (Abril et al. 2013). More importantly, Abril et al. (2013) found that tropical aquatic macrophytes export 80% of their GPP compared to just 36% for flooded forest. Therefore, the lack of a bespoke macrophyte PFT is indeed likely to be one reason for the discrepancy between our results and those of Borges, but largely due to their particularly high export efficiency to the river-floodplain network as opposed to differences in NPP. While being a significant limitation, creating and incorporating a macrophyte PFT would be a substantial undertaking given that the authors are unaware of any published dataset which has systematically mapped their distribution and abundance. It is important to note that while ORCHILEAK does not include the export of C from aquatic macrophytes it also neglects their NPP. Moreover, most aquatic macrophytes described in the literature have short (<1 year) life-cycles (Mitchel & Rogers, 1985). As such, while this model limitation is likely one of the causes for our relatively low estimate of riverine $CO_2$ evasion, it will only have a limited net effect on our estimate of the overall annual C balance (NBP, NEP) of the Congo basin."

L537-539: It is not correct that strong currents limit the abundance of aquatic macrophytes in the Amazon since most of their growth occurs on floodplains where they can cover large areas.

Ok. This is taken from previous literature (Borges et al., 2015, Scientific Reports) but we have removed this.

L570-572: Both these estimates of the % of NPP per year transferred to inland waters are based on the same model. What are the estimates for the Amazon based on empirical data? L572-582: This discussion of differences between the Amazon and Congo is too simplistic and not representative of the relevant conditions in either system. It would best be deleted unless considerable more information is added. Section 4.2: As noted above, it seems a real stretch to be projecting through the 21st century. L610-625: As this section is written as a comparison with Lauerwald et al. (submitted), it does seem suitable to include until Lauerwald et al is available. Also, there are publications that project hydrological and land use changes in the Amazon.

L570-582 have been removed.

Lauerwald et al (2020) has now been accepted for publication and so we have retained the comparison of the future projections for the Amazon and the Congo.

L626-624: This paragraph does not seem necessary since these systems are quite different from the Congo and other examples could be selected. Section 4.3: Lines 636-645 re-enforce the issues raised above regarding the projections through the 21st century and the question of whether their inclusion in this paper is warranted.

Conclusion L692-696: Is it likely that an increase in DOC from 9.5 to 11.5 mg C/L will cause ecologically meaningful changes in pH?

It is unclear so we have removed these sentences.

[revised manuscript text omitted]

---

## Editor Decision (ED1)

**ESD-2020-3: Editor Decision**

June 10$^{\text{th}}$, 2020

Dear Authors,

Following a fertile interaction discussion where pertinent concerns were raised and careful and concrete responses were put forward, the natural consequence most beneficial to the manuscript and to the audience that it addresses is clearly the diligent and thorough revision that already began to emerge from the discussions.

At this stage, additionally to all the relevant issues raised and debated in the interactive discussion stage, which are well motivated and discussed, I simply add a few supporting reflections are hereby shared in support and wrapping of the recent discussion.

It has become progressively clearer over the last two decades (even long before the studies from recent months) that uncertainties are consistently and vastly underestimated in earth system models (even beyond the scope of land surface processes) and that confidence runs excessively high in modelling efforts - both in forward and inverse modelling to be more precise.

There is a significant body of literature on the dynamics of model errors and uncertainty propagation in physics and in the geophysical sciences (for the sake of neutrality and to avoid expressing any preference, I will not give examples, but these can be found in geophysical and physical journals).

However, the relative disconnection between the communities most active in uncertainty modelling and those in more practical operational modelling has kept the latter mostly unaware of the most advanced developments in the dynamics of prediction errors and in uncertainty modelling at large.

Contributing to model errors are various factors ranging from formulations, parameterisations, initialisation, assimilation and calibration procedures, the narrow set of uncertainty propagation mechanisms (model structures are overly simplified and uncertainty propagation is often tied to such, so typically model errors are under-dispersive). The formulation of how such errors interact and propagate across the system is also found in the literature and actually provide helpful insights that assist in adjusting and improving model structures and data collection to cope with such problems.

As far as the present manuscript is concerned, it is still possible to preserve its scope wherever possible despite all of the above. While it is indeed important to make the inherent caveats of any modelling endeavour to the readers, it is also important to elicit the importance of such modelling investigations. Models will also be imperfect as they only capture a subset - if at all - of the system under analysis.

Using an inherently limited formulation that takes certain aspects of the dynamics into consideration has valid inspective and analytical merits that, even in an approximate or hypothetical setting, still contribute to formulate better hypothesis about key aspects in the system behaviour.

If we think of the ingenious stylised yet over-simplified model for convection that Lorenz put forward in the famous deterministic non-periodic flow paper in 1963 that is cited in endless papers about nonlinear dynamics in general and chaotic flows in particular, that model, albeit being admittedly physically unrealistic for its original intended purposes, was a genial system dynamic laboratory in itself. This is what models actually are: numerical laboratories, which provide important insights but at their core they do not need to be perfect if they are transparently taken for what they actually are.

Once the cards are clearly laid out on the table, and the scientific procedure is clearly explained for what it does represent and what it does not, for its merits and limitations, a scientific publication can still stand and be useful for the community.

This way, while recent developments cannot be ignored, the narrative of the manuscript can be aptly adjusted - especially with further analysis as suggested and within feasibility constraints - to incorporate such findings without entirely compromising the overall message. A message that indeed becomes more tentative and reflective of a modelling exercise to guide system interpretation and ignite scientific discussion rather than a factual proof of system behaviour.

All in all, it is crucial that the revised manuscript makes the workflow and results clear to the readers in a way that makes them understand that the study does not aim to state absolute facts but rather advance science by shedding new insights over relevant subsystem interactions in the earth system and especially on an area that really benefits from such studies given the scarcity of hard evidence to support stronger claims.

The authors are therefore encouraged to proceed with the intended revisions taking into careful consideration the concerns raised during the discussion stage, with the author responses already providing good promise in the right direction.

Thank you for your attention and best wishes.

Rui Perdigão

(ESD Editor)

---

## Author Response (AR2)

**We again sincerely thank both reviewers for taking the time to review and improve the paper. We hope that we satisfactorily address the various comments and questions below (responses in blue).**

**Reviewer 1:**

"GENERAL COMMENTS

The authors have addressed most of my previous comments. In the discussion of the differences between model results and data-driven estimates there could be a clarification in the comparison of numbers (see below). Also I do not agree with the way the authors dismiss differences in k to explain the divergence of results. Changes in k will lead to changes in emissions if the water is substantially over-saturated in CO2 (the change of k will have a small effect of the flux, in the water is very close to saturation, which is not the case of the Congo). I suggest that the authors provide a sensitivity analysis and increase the k by 50% and see how much the flux changes in the Congo, rather than refer to a study in the Amazon.

MAJOR COMMENTS

L78 : Battin et al (2009) report on aquatic heterotrophy (respiration) not C transfer to LOAC

Ok thanks, we realised that the transfer to LOAC quoted in Battin et al. (2009) is directly taken from Tranvik et al. (2009) so we have replaced former with the latter.

L 391 : The comparison of pCO2 in the main-stem Congo at Kisangani and Kinshasa does allow to conclude that the model reproduces the "broad spatial pattern of pCO2 measured in Borges et al. (2019)" as stated. There are other important features of "spatial pattern" of pCO2 observations such as much higher pCO2 values in the small and major tributaries in some cases with pCO2 values up to 18,000 ppm. I suggest to rephrase to "broad spatial pattern of pCO2 measured in the main-stem Congo reported by Borges et al. (2019)"

This is a fair point, we have rephrased as you suggested.

L409 : Borges et al. (2019) (Figure 20) also reports a 2 yr time series (2017-2018) of pCO2 at Kisangani. Wang et al. (2013) report a 1 yr time series of pCO2 at Kinshasa. It would make a convincing case if the time-series comparison was extended to these 2 other sites were complete annual cycles are available rather than only to Bangui on the basis that is the "most complete" as stated (whatever "complete" means in this context).

Thanks for these suggestions. We have tried to access the time series from both Borges et al. (2019) and Wang et al. (2013) but have been unable to access these publicly. Borges et al. (2019) uploaded two spreadsheets publicly detailing both their continuous and discrete measurements of pCO2 (https://zenodo.org/record/3413449#.XYm2eUYzaUk) but neither spreadsheet contains the time series at Kisangani. Nor does the supplementary data from Borges et al. (2015) contain a full time series.

If necessary, we would be happy to contact the authors of these studies directly but would require a longer extension and even then, we cannot be sure whether we would get access to these datasets.
We have at least been able to compare values for December and June at Kisangani (see Table 3).

L 584 : The value of k of 3.5 m/d is ok for high order streams (for instance minstem) and k used by Borges et al (2019) strongly increases in low order streams up to 39 m/d in order 1 streams. The largest fraction of CO2 emission reported by Borges et al. (2019) from the river network is related to low order streams.

L 587 : The statement that CO2 flux in orchileak is not sensitive to the k value is really strange and might require a few words of explanation. For a water pCO2 of 2800 ppm (seems to be sort of simulated pCO2 by the model in the Congo river) and a k of 3.5 m/d, the emission is more or less 287 mmol/m2/d. If the k is increased by 50% (=5.3 m/d) the resulting flux is more or less 430 mmol/m2/d, so also 50% higher.

So whether the fluxes are derived from ORCHILEAK or "data-driven models", a substantial increase of k leads to a substantial increase of k. I suggest that the authors provide a sensitivity analysis and increase the k by 50% in Orchileak and see how much the flux changes in the Congo, rather than refer to a study in the Amazon.

Based on the reported values of flux, k and surface area, I compute that the spatially integrated average of river pCO2 from ORCHILEAK is approximately 2800 ppm. The spatially integrated average of river pCO2 from Borges et al (2019) is approximately 5560 ppm based on numbers in Table 1. The average k Borges et al (2019) is 8 m/d, also based on data from Table 1,which is higher than the k value in ORCHILEAK of 3.5 m/d. So the difference between both estimates is related to both lower pCO2 and lower k in Orchileak. The authors should clarify how K is computed in Orchileak and discuss in light of the k computation scheme of Borges et al. (2019) that is transparently explained in the methods and supplements. This might shed some light on why the k values are so different in both studies. The higher pCO2 values in Borges et al. (2019) might result from not representing macrophytes and/or because the model under-estimates pCO2 in low order streams. This is not possible to check since the model results are presented as overall means and not information is given as function of stream size. I suggest that the authors present in more detail the pCO2 values simulated to tributaries and in particular low or der streams.

While possible, we would maintain that a bespoke sensitivity analysis of $k$ for the Congo is unnecessary. As shown in the sensitivity analysis of Lauerwald et al. (2017), $F$CO$_2$ in ORCHILEAK (as a process based/ physical approach) is not sensitive to $k$ due to the following:
- $F$CO$_2$ at a particular 6-min timestep is indeed calculated based on the water-atmosphere $p$CO$_2$ gradient, the water surface area and $k$ (see equations 76 and 77 of Lauerwald et al., 2017). As outlined in the methods section, in ORCHILEAK fixed $k$ values of 3.5 m d$^{-1}$ and 0.65 m d$^{-1}$ respectively are used for rivers (including open floodplains) and forested floodplains, the former similar to the 2.9 m d$^{-1}$ for rivers used by Borges et al. (2015[a]).

-In turn, at each time-step $pCO_2$ in the water column is calculated from the concentration of dissolved $CO_2$ and the temperature-dependent solubility of carbon (see equation 70). The concentration of dissolved $CO_2$ in turn depends on the input and decomposition of DOC (in situ production of $CO_2$) and the input of dissolved $CO_2$ from soils, litter and root respiration (see various equations from 58-69) on the input side, and the velocity $k$ that controls how quickly these $CO_2$ inputs can diffuse to the atmosphere.

-In ORCHILEAK, $k$ does have an important impact on $pCO_2$; i.e. a lower $k$ value will increase $pCO_2$, but this will also lead to a steeper water-air $CO_2$ gradient and so ultimately to approximately the same $FCO_2$ over time. In other words, over the scales covered in this research (the large catchment area and water residence times of the Congo), $FCO_2$ is ultimately mainly controlled by the allochthonous inputs of carbon to the river network, because by far the largest fraction of these C inputs is leaving the system via $CO_2$ emission to the atmosphere (as opposed to being laterally transferred downstream). The Cuvette Centrale is a hotspot region for $FCO_2$ (see Figure 8) due to the high allochthonous inputs of C to the river network, not due to particularly high or low $k$ values.

As a process-based model, ORCHILEAK represents directly the sources of C to the river network, and these are the main drivers of $CO_2$ emissions. In empirical studies, on the contrary, you don't know the C sources to the river with which to constrain $CO_2$ emissions; what is measured is the $pCO_2$, and you have to estimate the $k$ that has led to this $pCO_2$ under an unknown $CO_2$ input/production.

We absolutely take your point that further explanation is required and in line with the above, have changed the text in the manuscript as follows (Lines 590-610):

"One potential cause for the differences could be the river gas exchange velocity $k$. We applied a mean riverine gas exchange velocity $k_{600}$ of 3.5 m $d^{-1}$ which is similar to the 2.9 m $d^{-1}$ used by Borges et al. (2015[a]) but substantially smaller than the mean of approximately 8 m $d^{-1}$ estimated across Strahler orders 1-10 in Borges et al. (2019) (taking the contributing water surface area of each Strahler order into account). A sensitivity analysis was performed in Lauerwald et al. (2017) which showed that in the physical approach of ORCHILEAK, $CO_2$ evasion is not very sensitive to the $k$ value, unlike data-driven models. Namely, Lauerwald et al (2017) showed that an increase or decrease of $k_{600}$ for rivers and swamps (flooded forests) of 50% only led to 1% and -4% change in total $CO_2$ evasion, respectively. In ORCHILEAK, $k$ does have an important impact on $pCO_2$; i.e. a lower $k$ value will increase $pCO_2$, but this will also lead to a steeper water-air $CO_2$ gradient and so ultimately to approximately the same $FCO_2$ over time. In other words, over the scales covered in this research (the large catchment area and water residence times of the Congo), $FCO_2$ is mainly controlled by the allochthonous inputs of carbon to the river network, because by far the largest fraction of these C inputs is leaving the system via $CO_2$ emission to the atmosphere (as opposed to being laterally transferred downstream). Therefore, we do not consider $k$ to be a major source of the discrepancy. Additionally, our $k_{600}$ value of 0.65 m $d^{-1}$ for forested floodplains (based on Richey et al., 2002) compares well to recent a study which directly measured $k_{600}$ on two different flooded forest sites in the Amazon basin, observing a range of 0.24 to 1.2 m $d^{-1}$ (MacIntyre et al., 2019)."

L 578 comparison with Lauerwald et al. (2015) also suggests that you « substantially underestimate total riverine CO2 evasion".

This is true, though I think this is already acknowledged in the preceding sentence "but smaller than the 59.7 Tg C yr$^{-1}$ calculated by Lauerwald et al. (2015) and far smaller than that of Borges et al. (2015[a]), 133-177 Tg C yr$^{-1}$ or Borges et al. (2019), 251±46 Tg C yr$^{-1}$."

Regarding historical changes in LOAC fluxes, the authors might consider including in the discussion the recent paper of Moukandi N'kaya et al. (2020) that attribute decadal changes in DOC export from the Congo to changes in hydrology and inundation patterns.

Thank you for pointing out this paper. We have added a new paragraph to the discussion as follows (721-736):

"With these limitations in our understanding of tropical forest ecosystems in mind, over the entire simulation period (1861-2099) we estimate that aquatic $CO_2$ evasion will increase by 79% and the export of C to the coast by 67%. While, there are no long-term observations of aquatic $CO_2$ evasion in the Congo, a recent paper examined trends in observed DOC fluxes in the Congo at Brazzaville/Kinshasa over the last 30 years (Moukandi N'kaya et al. 2020). They found a 45% increase in the annual flux of DOC from 11.1 Tg C yr$^{-1}$ (mean from 1987-1993) to 16.1 Tg C yr$^{-1}$ (mean from 2006-2017). Comparing the same two periods, we find a smaller increase of 15% from 12.3 Tg C yr$^{-1}$ to 14.2 Tg C yr$^{-1}$. While our increase is substantially smaller, these observations are still over relatively short time scales and thus interannual variations could have considerable influence over the means of the two periods. Irrespectively it is encouraging that observations concur with the overall simulated increasing trend. Perhaps most interesting is that Moukandi N'kaya et al. (2020) attribute this increase to hydrological changes and specifically an increase in flood events in the central basin (including the Cuvette Centrale). Over this period, we too attribute the increase in carbon fluxes to the coast in part to climate change (Fig. 11 d) and over the full simulation period, the largest increase in DOC and $CO_2$ leaching into the aquatic system occurs within the Cuvette Centrale (Fig. A1)."

REFS

Moukandi N'kaya et al. (2020) Temporal Variability of Sediments, Dissolved Solids and Dissolved Organic Matter Fluxes in the Congo River at Brazzaville/Kinshasa, Geosciences 2020, 10, 341; doi:10.3390/geosciences10090341

Wang, Z. A., D. J. Bienvenu, P. J. Mann, K. A. Hoering, J. R. Poulsen, R. G. M. Spencer, and R. M. Holmes (2013), Inorganic carbon speciation and fluxes in the Congo River, Geophys. Res. Lett., 40, doi:10.1002/grl.50160"

**Reviewer 2:**

"Hastie and co-authors have responded well to comments on their initial submission. Though their projections of changes through this century are more uncertain and speculative than as suggested in the abstract, the main text does acknowledge the considerable problems with the projections. The following specific comments are minor or can be readily clarified.
Introduction
L49: Change 'sparcity' to 'paucity'.

Thanks, changed as suggested.

Methods
L140-142: 'Fixed gas exchange velocities of 3.5 m d-1 and 0.65 m d-1 respectively. are used for rivers (including open floodplains) and forested floodplains.'
A couple of references for these gas exchange velocities could be added. In particular, MacIntyre et al (2019) provides results for forested floodplains.
MacIntyre et al. 2019. Turbulence and gas transfer velocities in sheltered flooded forests of the Amazon basin. Geophysical Research Letters. doi.org/10.1029/2019GL083948

Thanks for pointing out this additional reference. MacIntyre et al. (2019) measured gas transfer velocities on two flooded forest sites in the Amazon ranging from 0.24 to 1.2 m d-1 so our value of 0.65 m d-1 lies in the middle of this range.

We have modified the text (lines 590-610) as follows (also to address the comments of Reviewer 1):
"One potential cause for the differences could be the river gas exchange velocity $k$. We applied a mean riverine gas exchange velocity $k_{600}$ of 3.5 m d$^{-1}$ which is similar to the 2.9 m d$^{-1}$ used by Borges et al. (2015[a]) but substantially smaller than the mean of approximately 8 m d$^{-1}$ estimated across Strahler orders 1-10 in Borges et al. (2019) (taking the contributing water surface area of each Strahler order into account). A sensitivity analysis was performed in Lauerwald et al. (2017) which showed that in the physical approach of ORCHILEAK, $CO_2$ evasion is not very sensitive to the $k$ value, unlike data-driven models. Namely, Lauerwald et al (2017) showed that an increase or decrease of $k_{600}$ for rivers and swamps (flooded forests) of 50% only led to 1% and -4% change in total $CO_2$ evasion, respectively. In ORCHILEAK, $k$ does have an important impact on $pCO_2$; i.e. a lower $k$ value will increase $pCO_2$, but this will also lead to a steeper water-air $CO_2$ gradient and so ultimately to approximately the same $FCO_2$ over time. In other words, over the scales covered in this research (the large catchment area and water residence times of the Congo), $FCO_2$ is mainly controlled by the allochthonous inputs of carbon to the river network, because by far the largest fraction of these C inputs is leaving the system via $CO_2$ emission to the atmosphere (as opposed to being laterally transferred downstream). Therefore, we do not consider $k$ to be a major source of the discrepancy. Additionally, our $k_{600}$ value of 0.65 m d$^{-1}$ for forested floodplains (based on Richey et al., 2002) compares well to recent a study which directly measured $k_{600}$ on two different flooded forest sites in the Amazon basin, observing a range of 0.24 to 1.2 m d$^{-1}$ (MacIntyre et al., 2019)."

Figure 3a: Are fluxes from L. Tanganyika and other lakes included? If so, what is the source of the gas concentrations and fluxes.

No, as lakes are not directly represented in ORCHILEAK.

L241-242: 'The best performing climate forcing dataset was ISIMIP2b followed by Princeton GPCC with root mean square errors (RMSE) of 29% and 40% and Nash Sutcliffe efficiencies (NSE) of 0.20 and -0.25, respectively.'
Are RMSE and NSE values considered fair or good?

As the other reviewer previously pointed out, I suppose this is for the reader to decide. Also, these values are before calibration (the metrics for ISIMIP improve after calibration).

248-250: 'water residence times - 0.5 (days) for floodplain reservoirs'
What is the basis for residence times of 0.5 days on the floodplains, as it seems too fast?

The 0.5 days actually refers to tau_flood ($\tau_{flood}$), a parameter which helps to control the residence time. However, this parameter is multiplied by a topographical index and the flooded fraction of the grid cell to calculate residence time, and residence time is thus changing at each time step but is not explicitly calculated as diagnostic output variable . The $\tau_{flood}$ value of 0.5 was arrived at by calibrating against two very different rivers (the Congo and the Oubangui) and flooded seasonality (GIEMS), as well as trying to represent a large and diverse basin.

We apologise for the mistake in the manuscript and the confusion caused. We have changed the text as follows (247-256):

"For ISIMIP2b we further calibrated key hydrological model parameters, namely the constants (tau, $\tau$) which help to control the water residence time of the groundwater (=slow reservoir), headwaters (= fast reservoir) and floodplain reservoirs in order to improve the simulation of observed discharge at Brazzaville and Oubangui (Table 2). To do so, we tested different combinations of $\tau$ values for the three reservoirs, eventually settling on 1, 0.5 and 0.5 (days) for the slow, fast and floodplain reservoirs respectively, all three being reduced compared to those values used in the original ORCHILEAK calibration for the Amazon (Lauerwald at al., 2017). The actual residence time of each reservoir is calculated at each time step. The residence time of the flooded reservoir for example, is a product of $\tau_{flood}$, a topographical index and the flooded fraction of the grid cell."

L255-257:' As in previous studies on the Amazon basin (Lauerwald et al. 2017, Hastie et al., 2019) we defined bank-full discharge, i.e. the threshold discharge at which floodplain inundation starts (i.e. overtopping of banks), as the median discharge (50th percentile) of the present-day climate forcing period (1990 to 2005).'

The response to the comment on the initial submission ('The concept of bank-full discharge as a threshold for initiation of inundation of floodplains is questionable as applied to tropical floodplain such as those in the Amazon or Congo. Studies inundation dynamics in the Amazon with detailed measurements or modeling indicate that inundation occurs more or less continuously as the rivers rise and that the water comes from both the rivers and uplands.') was mis-interpreted. The issue being raised was with regard to the proportion of water from different sources. The issue concerns the observation that natural floodplains, such as those in the Amazon and Congo, have channels that connect the rivers to the floodplains such that waters rise on the floodplains in concert with the river rise, not just after bank-full discharge is reached.

Ok, we understand now. In regards to this, ORCHILEAK simulates both precipitation onto the floodplain and evaporation from the floodplain. This precipitation in turn feeds directly into the floodplain reservoir and thus in ORCHILEAK inundation on the floodplain is not only a result of overtopping but also local precipitation directly onto the floodplain.

Table 1: Is there snowfall in basin?

No, I don't think there is, good point. We have deleted reference to this in the Table

How does the river area used compare to the recent estimate by Allen and Pavelsky (2018. Global extend of rivers and streams. Science 361: 585–588) ?

According to their summary shapefile per basin (downloaded here- https://drive.google.com/file/d/11hzVVg6OEs1c7zIKjuy0u4WeE0UG6BsH/view) Allen and Pavelsky estimate a total river and stream surface area of 17,903 km2 for the Congo basin, which falls at the lower end when compared to existing estimates such as 23,670 km$^2$ from Borges et al. (2019) and 26,517 km2 from Borges et al. (2015, based on Raymond et al., 2013). While this may indicate that both our estimate and that of Borges et al. (2019) are too high, that is not the reason for the discrepancy between our FCO2 and Borges' given that our estimates are relatively similar.

We have added a sentence on this (611-616):

"Another potential reason for our smaller riverine $CO_2$ evasion could be river surface area. We simulate a mean present day (1980-2010) total river surface area of 25,900 km$^2$, compared to the value of 23,670 km$^2$ used in Borges et al (2019, supplementary information) and so similarly we think that this can be discounted as a major source of discrepancy. However, it should be noted that both estimates are high compared to the recent estimate of 17,903 km$^2$ based on analysis of Landsat images (Allen & Pavelsky, 2018)."

Results
Table 2: State 'observed flooded area' is from GIEMS. As noted correctly in the text (L364-379), the GIEMS data under-estimate the flooded area.

Changed as suggested: "Observed flooded area is from GIEMS (Papa et al., 2010, Becker et al., 2018)."

L448-449: 'We simulate a mean annual flux of DOC throughfall from the canopy of 27 ±1 Tg C yr-1. How does this compare to measured fluxes (e.g. Filoso et al. 1999. Composition and deposition of throughfall in a flooded forest archipeligo (Anavilhanas, Negro River, Brazil). Biogeochemistry 45:169-195)."

This paper (Filoso et al., 1999) was indeed used in the ORCHILEAK model development paper (Lauerwald et al., 2017) for validation of throughfall and it compared well. Please see Figure 11 and Table 2 of Lauerwald et al., 2017.

[revised manuscript text omitted]